# 🧑‍💻 GUI-Actor: Coordinate-Free Visual Grounding for GUI Agents

Qianhui Wu[*1]   Kanzhi Cheng[*2]   Rui Yang[*3]   Chaoyun Zhang[1]   Jianwei Yang[1]

Huiqiang Jiang[1]   Jian Mu[2]   Baolin Peng[1]   Bo Qiao[1]   Reuben Tan[1]   Si Qin[1]   Lars Liden[1]

Qingwei Lin[1]   Huan Zhang[3]   Tong Zhang[3]   Jianbing Zhang[2]   Dongmei Zhang[1]   Jianfeng Gao[†1]

[1]Microsoft       [2]Nanjing University       [3]University of Illinois Urbana-Champaign

## Abstract

One of the principal challenges in building VLM-powered GUI agents is visual grounding—localizing the appropriate screen region for action execution based on both the visual content and the textual plans. Most existing work formulates this as a text-based coordinate generation task. However, these approaches suffer from several limitations: weak spatial-semantic alignment due to lack of explicit spatial supervision; inability to handle ambiguous supervision targets, as single-point predictions penalize valid variations; and a mismatch between the dense nature of screen coordinates and the coarse, patch-level granularity of visual features extracted by models like Vision Transformers. In this paper, we propose `GUI-Actor`, a VLM-based method for coordinate-free GUI grounding. At its core, `GUI-Actor` introduces an attention-based action head that learns to align a dedicated `<ACTOR>` token with all relevant visual patch tokens, enabling the model to propose one or more action regions in a single forward pass. In line with this, we further design a grounding verifier to evaluate and select the most plausible action region from the candidates proposed for action execution. Extensive experiments show that `GUI-Actor` outperforms prior state-of-the-art methods on multiple GUI action grounding benchmarks, with improved generalization to unseen screen resolutions and layouts. Notably, **GUI-Actor-7B** achieves scores of **40.7** with Qwen2-VL and **44.6** with Qwen2.5-VL as backbones, outperforming **UI-TARS-72B** (**38.1**) on ScreenSpot-Pro, with significantly fewer parameters and training data. Furthermore, by incorporating the verifier, we find that fine-tuning only the newly introduced action head (∼100M parameters for 7B model) while keeping the VLM backbone frozen is sufficient to achieve performance comparable to previous state-of-the-art models, highlighting that `GUI-Actor` can endow the underlying VLM with effective grounding capabilities without compromising its general-purpose strengths. Project page: `https://aka.ms/GUI-Actor`.

## 1   Introduction

With the rapid advancement of large language models (LLMs) and vision-language models (VLMs), there is increasing interest in building GUI (Graphical User Interface) agents that understand natural language instructions and autonomously interact with software interfaces across platforms such as desktops [1, 2], mobile devices [3], and web applications [4]. Effective GUI agents require two core capabilities: (i) multimodal perception to interpret visual and linguistic cues, and (ii) action

---

[*]Equal contribution: qianhuiwu@microsoft.com, chengkz@smail.nju.edu.cn, ry21@illinois.edu.
[†]Leadership.

39th Conference on Neural Information Processing Systems (NeurIPS 2025).

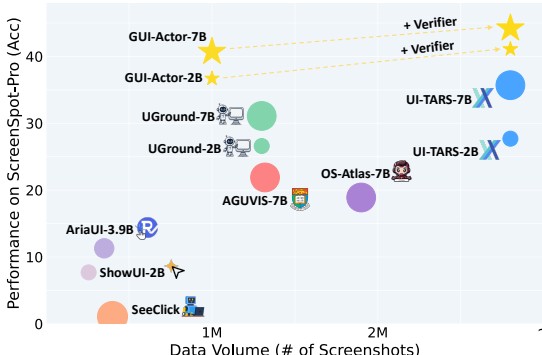 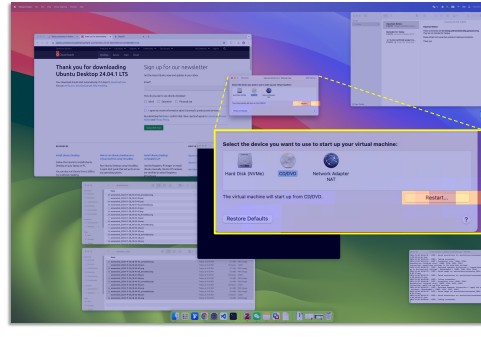

Instruction: Restart from CDs

Figure 1: **Left**: Model performance vs. training data scale on the ScreenSpot-Pro benchmark. Higher and more left is better; larger points indicate models with more parameters. We only show GUI-Actor models built upon Qwen2-VL here for fair comparison. With Qwen2.5-VL as the backbone, GUI-Actor-3B/7B reaches scores up to 42.2/44.6 (without Verifier). **Right**: Illustration of action attention. GUI-Actor grounds target elements by attending to the most relevant visual regions.

execution to interact with digital environments via mouse, keyboard, or touchscreen [5, 6]. While early systems relied on structured metadata (*e.g.* HTML, DOM trees, or view hierarchies) [1], such data is often noisy, inconsistent, or unavailable across platforms. Recent work thus emphasizes visual GUI agents that perceive interfaces directly from rendered screenshots, akin to human users [7]. A central challenge in this paradigm is *visual grounding*: mapping natural language plans to screen regions. Most existing methods treat this as a coordinate generation task, producing screen positions (*e.g.* "x=0.125, y=0.23") through the same text generation mechanisms used by LLMs [8].

However, representing GUI actions through coordinate generation, where models output screen positions as text tokens (*e.g.*, x=..., y=...) introduces several intrinsic limitations. First, *spatial-semantic alignment is weak*: generating discrete coordinate tokens requires the model to implicitly map visual inputs to numeric outputs via a language modeling head, without any explicit spatial inductive bias. This process is inefficient, data-intensive, and prone to errors due to the lack of direct supervision linking visual features to action locations. Second, *supervision signals are ambiguous*: many GUI actions, such as clicking within a button, allow for a range of valid target positions. However, coordinate-based methods typically treat the task as single-point prediction, penalizing all deviations—even reasonable ones—and failing to capture the natural ambiguity of human interaction. Finally, there is a *granularity mismatch between vision and action space*: while coordinates are continuous and high-resolution, vision models like Vision Transformers (ViTs) [9] operate on patch-level features. This mismatch forces the model to infer dense, pixel-level actions from coarse visual tokens, which undermines generalization to diverse screen layouts and resolutions.

Although some recent approaches [10] attempt to enrich spatial grounding by predicting bounding boxes instead of single points, they still represent these regions as raw coordinate strings (*e.g.* x_min, y_min, x_max, y_max) that are detached from the visual features. Without architectural components such as ROI pooling [11] or spatial attention mechanisms [12], such methods fall short of bridging the gap between linguistic intent and grounded visual action.

Rethinking how humans interact with digital interfaces reveals a key insight: *humans do not calculate precise screen coordinates before acting—they perceive the target element and interact with it directly*. Motivated by this observation, we propose `GUI-Actor`, a VLM augmented with an attention-based action head, enabling coordinate-free visual grounding that more closely mimics human behavior. Unlike prior methods that treat action grounding as a coordinate prediction task, `GUI-Actor` learns to attend directly to relevant visual regions without relying on numeric coordinate generation. At the core of `GUI-Actor` is a dedicated `<ACTOR>` token, which encodes the grounding context by jointly processing visual input and natural language instructions. An attention mechanism then learns to align this token with the most relevant GUI regions by attending over visual patch tokens from the screenshot. The resulting attention map naturally identifies actionable regions on the interface.

To address the inherent ambiguity in GUI interactions, where multiple points within a UI element (*e.g.* a button) may all be valid, `GUI-Actor` is trained using multi-patch supervision. All visual

patches overlapping with ground-truth bounding boxes are labeled as positives, while others are treated as negatives. This supervision strategy allows the model to tolerate spatial ambiguity and reduces over-penalization of reasonable action variations. Furthermore, because `GUI-Actor` grounds actions directly at the vision backbone's native spatial resolution, it avoids the granularity mismatch of previous methods and generalizes more robustly across different screen sizes, resolutions, and layouts. Finally, to support decision refinement, we further enhance GUI-Actor by presenting a lightweight grounding verifier that evaluates multiple candidate regions and selects the most plausible one for action execution.

Our contribution can be summarized as follows:

1. We revisit recent coordinate generation-based approaches for visual grounding in GUI agents, identify their limitations—such as weak spatial-semantic alignment, ambiguous supervision targets, and mismatched feature granularity—and propose `GUI-Actor`, a novel coordinate-free method that effectively addresses these issues.

2. We design an attention-based action head, which can generate multiple candidate regions in a single forward pass, offering flexibility for downstream modules such as search strategies.

3. We introduce a grounding verifier to select the most likely action region among the candidates proposed from the action attention map. We show that this verifier can be easily integrated with other grounding methods for a further performance boost.

4. Extensive experiments demonstrate that `GUI-Actor` outperforms the state-of-the-art methods trained on a similar scale of data across multiple GUI action grounding benchmarks, and exhibits greater robustness to unseen screen sizes and resolutions. Remarkably, the 2B version of `GUI-Actor` even surpasses several competing 7B models. Furthermore, by leveraging the verifier, `GUI-Actor` with lightweight training (*i.e.*, freezing the backbone LLM and fine-tuning only the newly introduced ∼100M parameters in the action head) can effectively equip the underlying VLM with grounding capabilities without compromising its general-purpose strengths.

## 2 Related Work

**LLM/VLM-Powered GUI Agents.** The advent of LLMs and VLMs has catalyzed the development of GUI agents that can understand natural language instructions and perform complex tasks across mobile [13], web [14, 15], and desktop environments [16, 1, 2]. Early research focused on designing autonomous agent frameworks [4, 17, 18] that prompt commercial models to interact with operating systems via code generation [19, 20] or tool use [21, 22]. With rising demand for open-source and customizable agents, a parallel line of work focuses on training LLMs/VLMs for enhanced agentic capabilities, including GUI understanding, planning, and execution [7, 10, 23, 24]. The key to these efforts is collecting GUI-specific training data, such as OCR annotations [25], interface summaries [26], QA pairs [23], and large-scale task demonstrations [15, 27–32].

A central requirement of agent development is the ability to interact with realistic GUI environments deployed in virtual machines and Chrome-based browsers. While early agents operated over structured metadata like HTML or accessibility trees [14, 33], such representations are brittle and inconsistent across platforms [34, 7]. Consequently, recent trends have shifted toward a *vision-centric paradigm*, where agents interact with raw screenshots using mouse and keyboard inputs [35, 36], closely mimicking human behavior. Within this setting, a central challenge emerges: grounding natural language instructions to specific GUI regions, referring to as *GUI Visual Grounding*.

**GUI Visual Grounding.** Given a GUI screenshot and a natural language instruction, GUI visual grounding aims to locate the target region for interaction. Although conceptually related to grounding in natural images, this task presents unique challenges due to the semantic density and structural regularity of GUI layouts [34, 8]. A common approach frames GUI grounding as a text-based coordinate prediction task, where models generate point positions (*e.g.*, x=..., y=...) as output language tokens [37, 38]. This formulation has led to widespread adoption due to its simplicity and compatibility with existing LLMs/VLMs. To improve performance, prior works have scaled both models and training data [39–43, 10, 44]. UGround [8] proposes a data pipeline for synthesizing diverse GUI grounding examples, while OS-Atlas [45] offers a multi-platform dataset and a unified GUI action model. More recently, Xu et al. [46] introduced a training-free approach that performs GUI grounding by leveraging the internal attention of VLMs.

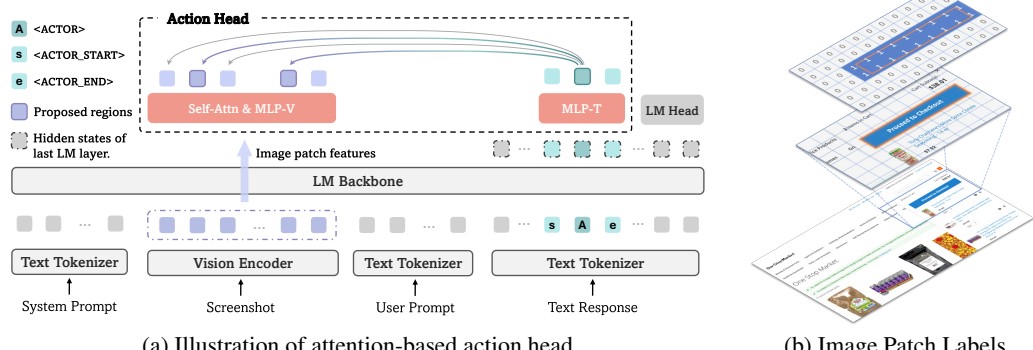

(a) Illustration of attention-based action head.      (b) Image Patch Labels

Figure 2: Overview of GUI-Actor. (a) Illustration of how the action head works with a VLM for coordinate-free GUI grounding. (b) Illustration of the spatial-aware multi-patch supervision for model training. We label all image patches that are partially or fully covered by the ground-truth bounding box as positive (1) and all others as negatives (0).

Despite their success, coordinate-based methods suffer from key limitations, including weak spatial inductive bias, ambiguous point supervision, and resolution mismatches between visual features and action targets. This paper presents a compelling alternative to the prevailing coordinate-based method: GUI-Actor, a novel *coordinate-free grounding framework* for GUI agents. It introduces an `<ACTOR>` token that attends directly to relevant image patches via an attention-based action head, enabling more human-like grounding while mitigating the limitation of coordinate-based methods.

## 3 The Design of GUI-Actor

Considering the limitations of text-based coordinate generation, *e.g.*, weak spatial-semantic alignment and ambiguous supervision targets, we draw inspiration from how humans interact with GUIs. Rather than computing precise coordinates, humans typically visually identify the intended element and then directly act on it, by tapping with a finger or positioning a mouse cursor. Motivated by this, `GUI-Actor` explores a novel architecture for GUI visual grounding: we first introduce a special token `<ACTOR>` as the contextual anchor, and then train an action attention head to attend from this anchor to image patches corresponding to the target element. Finally, we present a grounding verifier to select the most semantically appropriate target among the multiple candidates derived from the attention map.

`<ACTOR>` **Token as a Contextual Anchor**    Given a screenshot image $\mathcal{I}$ and an instruction $q$, coordinate generation based methods typically train the VLM to produce a sequence of $\{\boldsymbol{x}_{1:i-1}, \boldsymbol{x}_{i:i+m}, \boldsymbol{x}_{i+m+1:j-1}, \boldsymbol{x}_{j:j+n}, \boldsymbol{x}_{j+n+1:N}\}$, where $m, n > 0$, $N$ is the total length of the output sequence, $i > 1$, and $j - 1 > i + m + 1$. For example, in `pyautogui.click(x=0.123, y=0.234)` Xu et al. [7], the segments $\boldsymbol{x}_{i:i+m}$ and $\boldsymbol{x}_{j:j+n}$ correspond to the tokenized x- and y-coordinates, respectively. The segment $\boldsymbol{x}_{i+m+1:j-1}$ represents the separator between them, while the rest captures the surrounding context. In this work, we replace the coordinate span $\{\boldsymbol{x}_{i:i+m}, \boldsymbol{x}_{i+m+1:j-1}, \boldsymbol{x}_{j:j+n}\}$ with three special tokens to facilitate coordinate-free grounding and better context integration from both the visual input and textual instruction:

$$\text{VLM}(\mathcal{I}, q) = \{\boldsymbol{x}_{1:i-1}, \text{<ACTOR\_START>}, \text{<ACTOR>}, \text{<ACTOR\_END>}, \boldsymbol{x}_{i+3:N}\}. \tag{1}$$

We use the final-layer hidden state of `<ACTOR>`, *i.e.*, $h_{\text{<ACTOR>}}$, as a contextual anchor for computing action attention over the visual patch tokens.

**Attention-Based Action Head**    Let $v_1, \ldots, v_M$ denote the visual patch features extracted by the Vision Encoder of the VLM from the input screenshot, where each $v_i \in \mathbb{R}^d$. The action head computes an attention distribution from the `<ACTOR>` token over these visual patches to identify the target action region.

To incorporate GUI-aware contextual information, we first apply a self-attention layer over the visual patch features. This allows the model to aggregate semantics across spatially related patches, enabling patches that belong to the same GUI element to share coherent representations:

$$\tilde{v}_1, \ldots, \tilde{v}_M = \text{SelfAttn}(v_1, \ldots, v_M), \tag{2}$$

where $\tilde{v}_i \in \mathbb{R}^d$ denotes the contextualized feature for the $i$-th patch after self-attention module. Note that we do not introduce additional positional embeddings in this self-attention layer, as the input features from the ViT backbone already incorporate positional information.

Next, we project both the `<ACTOR>` token representation $h_{\texttt{<ACTOR>}}$ and the contextualized patch features $\tilde{v}_1, \ldots, \tilde{v}_M$ into a shared embedding space via two separate MLPs, and obtain $z$ and $z_i \in \mathbb{R}^d$:

$$z = \text{MLP}_\text{T}(h_{\texttt{<ACTOR>}}), \quad z_i = \text{MLP}_\text{V}(\tilde{v}_i). \tag{3}$$

Finally, we compute attention scores between the `<ACTOR>` token and each visual patch. Let $M$ denote the total number of image features that are input to the LM backbone, the resulting attention weights $a_1, \ldots, a_M$ form an attention map over the screen, indicating the most relevant region for grounding the action.

$$\alpha_i = \frac{z^\top z_i}{\sqrt{d}}, \quad a_i = \frac{\exp(\alpha_i)}{\sum_{j=1}^M \exp(\alpha_j)}, \quad \text{for } i = 1, \ldots, M. \tag{4}$$

**Spatial-Aware Multi-Patch Supervision**  A key advantage of our approach is the ability to leverage dense and spatially structured learning signals from bounding-box supervision. Rather than relying on a single, potentially ambiguous click point as in traditional coordinate-based methods, `GUI-Actor` treats all image patches that are partially or fully covered by the ground-truth bounding box as positive examples ($y_i$=1) and all others as negatives ($y_i = 0$), where $y_i$ denotes the label associated with $v_i$. This allows the model to more effectively capture the full spatial extent of actionable elements. An illustration is provided in Figure 2b. For more details, please refer to Appendix B.

We train the model using a combination of next-token prediction (NTP) loss and action attention loss:

$$\mathcal{L} = \mathcal{L}_\text{NTP} + \mathcal{L}_\text{Action\_Attn}, \tag{5}$$

The action attention loss is defined as:

$$\mathcal{L}_\text{Action\_Attn} = \sum_{i=1}^M p_i \log \frac{p_i}{a_i}, \quad p_i = \frac{y_i}{\sum_{j=1}^M y_j + \epsilon}, \ i = 1, \ldots, M, \tag{6}$$

where $\epsilon$ is a small constant for numerical stability.

## 4  Grounding Verifier

A key advantage of our attention based action grounding model is its ability to produce multiple candidate action regions in a single forward pass, without incurring additional inference cost. This is a natural consequence of the attention-based design, where the model assigns scores to all visual patches simultaneously. This efficiency creates an opportunity: rather than relying solely on the top-scoring patch, we can introduce a lightweight verification step to select the most semantically appropriate target among the candidates.

With the insight that verification is often easier than generation [47], we propose a *Grounding Verifier*, a lightweight VLM module which takes as input the instruction and a screenshot with a visual marker placed at the proposed location, and predicts whether the marked region correctly fulfills the task intent. This verifier serves as a decision refinement layer, allowing the system to reflect on its action before execution.

**Data & Training**  Training data for the verifier is constructed from the OS-Atlas dataset [45], which spans desktop, mobile, and web domains. This dataset provides triplets of the form (image, query, bounding box), where each image is associated with multiple queries and their corresponding bounding boxes. For each triplet, we generate a *positive example* by placing a visual

marker (*i.e.*, a hollow red circle) at the center of the bounding box, treating it as the correct grounding point for the given query. To create *negative examples*, we apply two strategies: (1) selecting the center of an incorrect bounding box from the same image; (2) randomly sampling a point outside the target region. Each resulting training instance is formatted as a tuple $(\mathcal{I}, \mathbf{x}, y)$, where $\mathcal{I}$ is an image with a marked point, $\mathbf{x}$ is the corresponding language instruction, and $y \in \{\text{'True'}, \text{'False'}\}$ is the ground-truth label indicating whether the point correctly satisfies the instruction. More details are deferred to Appendix F.1.

We fine-tune the verifier from a lightweight VLM using standard supervised learning. The model takes $(\mathcal{I}, \mathbf{x})$ as input and is trained to generate the correct token $y$. The training objective is the cross-entropy loss:

$$\mathcal{L}_{\text{Verifier}} = -\log P_{\theta_v}(y \mid \mathbf{I}, \mathbf{x}),$$

where $P_{\theta_v}$ denotes the output probability from the verifier model with parameters $\theta_v$.

**Inference**    At inference time, `GUI-Actor` predicts the final action location by combining natural language generation with visual grounding. Given the current GUI state and a user instruction, `GUI-Actor` first generates an agent response via standard decoding, for example, producing a string like `pyautogui.click(<ACTOR_START><ACTOR><ACTOR_END>)` that includes the special `<ACTOR>` token. We then extract the hidden state corresponding to `<ACTOR>` and use the action head to compute attention over all visual patches. This attention distribution serves as a spatial activation map, identifying the most relevant screen region for executing the predicted action.

To identify the most semantically appropriate region among the top-$K$ attention-weighted patches, we use the verifier $\theta_v$ to score each candidate by marking it on the image $\mathcal{I}$ and evaluating its alignment with the instruction $x$. For each marked image, the verifier outputs probabilities for `'True'` and `'False'` tokens, and we define the selection score as:

$$\mathbf{s}(\mathcal{I}, x) = \frac{P_{\theta_v}(\text{'True'} \mid \mathcal{I}, x)}{P_{\theta_v}(\text{'True'} \mid \mathcal{I}, x) + P_{\theta_v}(\text{'False'} \mid \mathcal{I}, x)}. \tag{7}$$

Candidates are evaluated in descending order of attention weights, and we return the first one exceeding a confidence threshold (e.g., $\mathbf{s}(\mathcal{I}, x) > \gamma$) without further evaluation.

## 5   Experiments

**Implementation Details**    We implement `GUI-Actor` using PyTorch and Huggingface Transformers. Unless otherwise specified, we adopt Qwen-2-VL-7B-Instruct [38] as the backbone VLM for both `GUI-Actor` and the baselines to ensure a fair comparison with previous state-of-the-art methods. For the re-implementation of baseline Aguvis-7B with both point supervision *(point sup.)* and bounding-box supervision *(bbox sup.)*, we directly use the official source code provided by Aguvis [7]. The number of attention heads in the self-attention layer is set to 8; Both MLP components are two-layer feedforward networks with a GELU activation in between. We use the same dimensionality as the backbone VLM for all configurations of the action head. The grounding verifier is finetuned from UI-TARS-2B-SFT [10]. During inference, we construct a pool of $K = 20$ candidates and apply a confidence threshold of $\gamma = 0.95$ for tasks in ScreenSpot-Pro and $\gamma = 0.8$ for ScreenSpot and ScreenSpot-v2. Following Aguvis [7], we structure our training data as sequences of `pyautogui`-style operations, but replace the original coordinates with the special tokens, as described in Section 3. Our full training recipe is built from several public GUI datasets, comprising $\sim$1M screenshots. Both `GUI-Actor` and the two baseline models are trained using the data recipe summarized in Table 7 for 1 epoch. Additional dataset details are provided in Appendix D. To train `GUI-Actor`, we begin by freezing all backbone VLM parameters and training only the newly introduced components of the action head ($\sim$20M/$\sim$100M parameters for 2B/7B backbone). After this warm-up phase, we finetune the entire model using standard supervised learning.

**Evaluation Benchmarks & Metric**    We evaluate `GUI-Actor` and other baseline methods on three well-established GUI visual grounding benchmarks: ScreenSpot [34], ScreenSpot-v2 [45], and ScreenSpot-Pro [48], with the last featuring higher-resolution interfaces and greater domain shift (e.g., industrial software, multi-window layouts), serving as a practical testbed for generalization. For the evaluation metric, we use *Element Accuracy*, which measures the proportion of predictions

Table 1: Performance comparison on *ScreenSpot-Pro*, which features higher-resolution interfaces and greater domain shift (*e.g.*, industrial software, multi-window layouts), serving as a practical testbed for generalization.

| | Dev | Creative | CAD | Scientific | Office | OS | Avg-Text | Avg-Icon | Avg |
|---|---|---|---|---|---|---|---|---|---|
| GPT-4o | 0.7 | 0.6 | 1.5 | 1.2 | 0.9 | 0 | 1.3 | 0 | 0.8 |
| Claude Compute | 12.6 | 16.8 | 11.9 | 25.8 | 26.9 | 8.1 | 23.4 | 7.1 | 17.1 |
| OS-Atlas-4B | 3.7 | 2.3 | 1.5 | 7.5 | 4.8 | 3.1 | 5.0 | 1.7 | 3.7 |
| ShowUI-2B | 9.4 | 5.3 | 1.9 | 10.6 | 13.5 | 6.6 | 10.8 | 2.6 | 7.7 |
| UGround-V1-2B | 27.4 | 26.7 | 14.6 | 34.3 | 38.3 | 17.9 | - | - | 26.6 |
| UI-TARS-2B | 26.4 | 27.6 | 14.6 | 39.8 | 42.6 | 14.3 | 39.6 | 8.4 | 27.7 |
| **GUI-Actor-2B** | 34.8 | 35.5 | 28.4 | 38.2 | 53.9 | 30.6 | 50.7 | 14.1 | **36.7** |
| **GUI-Actor-2B + Verifier** | 41.8 | 36.7 | 34.5 | 41.3 | 62.6 | 36.2 | 57.6 | 16.1 | **41.8** |
| Qwen2-VL-7B | 1.3 | 0.9 | 0.4 | 3.5 | 3.0 | 0.5 | 2.5 | 0.2 | 1.6 |
| SeeClick-9.6B | 0.3 | 0.6 | 1.9 | 2.0 | 0.9 | 1.5 | 1.8 | 0 | 1.1 |
| Aria-UI-2-5.3B | 8.4 | 14.7 | 6.1 | 18.1 | 16.1 | 2.6 | 17.1 | 2.0 | 11.3 |
| OS-Atlas-7B | 17.7 | 17.9 | 10.3 | 24.4 | 27.4 | 16.8 | 28.1 | 4.0 | 18.9 |
| Aguvis-7B | 16.1 | 21.4 | 13.8 | 34.6 | 34.3 | 19.4 | - | - | 22.9 |
| UGround-V1-7B | 28.1 | 31.7 | 14.6 | 39 | 49.6 | 24.5 | - | - | 31.1 |
| UI-TARS-7B | 36.1 | 32.8 | 18.0 | 50.0 | 53.5 | 24.5 | 47.8 | 16.2 | 35.7 |
| **GUI-Actor-7B** | 38.8 | 40.2 | 29.5 | 44.5 | 56.5 | 36.2 | 55.8 | 16.4 | **40.7** |
| **GUI-Actor-7B + Verifier** | 38.8 | 40.5 | 37.2 | 44.5 | 64.8 | 43.9 | 60.7 | 17.6 | **44.2** |
| CogAgent-18B | 8.0 | 5.6 | 6.1 | 13.4 | 10.0 | 3.1 | 12.0 | 0.8 | 7.7 |
| UGround-72B-V1 | 31.1 | 35.8 | 13.8 | 50.0 | 51.3 | 25.5 | - | - | 34.5 |
| UI-TARS-72B | 40.8 | 39.6 | 17.2 | 45.7 | 54.8 | 30.1 | 50.9 | 17.5 | 38.1 |

Table 2: Performance comparison on *ScreenSpot*.

| | Mobile-Text | Mobile-Icon | Desktop-Text | Desktop-Icon | Web-Text | Web-Icon | Avg |
|---|---|---|---|---|---|---|---|
| GPT-4 | 22.6 | 24.5 | 20.2 | 11.8 | 9.2 | 8.8 | 16.2 |
| GPT-4o | 20.2 | 24.9 | 21.1 | 23.6 | 12.2 | 7.8 | 18.3 |
| Claude Computer Use | - | - | - | - | - | - | 83.0 |
| Gemini 2.0 | - | - | - | - | - | - | 84.0 |
| UGround-V1-2B | 89.4 | 72.0 | 88.7 | 65.7 | 81.3 | 68.9 | 77.7 |
| UI-TARS-2B | 93.0 | 75.5 | 90.7 | 68.6 | 84.3 | 74.8 | 82.3 |
| **GUI-Actor-2B** | 93.0 | 79.9 | 88.1 | 78.6 | 90.9 | 84.0 | **86.5** |
| **GUI-Actor-2B + Verifier** | 93.8 | 81.2 | 89.7 | 80.7 | 91.3 | 80.6 | **86.9** |
| Qwen2-VL-7B | 75.5 | 60.7 | 76.3 | 54.3 | 35.2 | 25.7 | 55.3 |
| CogAgent-7B | 67.0 | 24.0 | 74.2 | 20.0 | 70.4 | 28.6 | 47.4 |
| SeeClick-9.6B | 78.0 | 52.0 | 72.2 | 30.0 | 55.7 | 32.5 | 53.4 |
| Magma-8B | 60.4 | 58.5 | 75.3 | 52.9 | 69.1 | 52.0 | 60.3 |
| Aguvis-G-7B | 88.3 | 78.2 | 88.1 | 70.7 | 85.7 | 74.8 | 81.8 |
| OS-Atlas-7B | 93.0 | 72.9 | 91.8 | 62.9 | 90.9 | 74.3 | 82.5 |
| Aguvis-7B | 95.6 | 77.7 | 93.8 | 67.1 | 88.3 | 75.2 | 84.4 |
| UGround-v1-7B | 93.0 | 79.9 | 93.8 | 76.4 | 90.9 | 84.0 | 86.3 |
| UI-TARS-7B | 94.5 | 85.2 | 95.9 | 85.7 | 90.0 | 83.5 | 89.5 |
| **GUI-Actor-7B** | 94.9 | 82.1 | 91.8 | 80.0 | 91.3 | 85.4 | **88.3** |
| **GUI-Actor-7B + Verifier** | 96.0 | 83.0 | 93.8 | 82.1 | 92.2 | 87.4 | **89.7** |
| UI-TARS-72B | 94.9 | 82.5 | 89.7 | 88.6 | 88.7 | 85.0 | 88.4 |
| Aguvis-72B | 94.5 | 85.2 | 95.4 | 77.9 | 91.3 | 85.9 | 89.2 |
| UGround-V1-72B | 94.1 | 83.4 | 94.9 | 85.7 | 90.4 | 87.9 | 89.4 |

where the click point falls within the ground-truth bounding box of the target element. Please refer to Appendix E for more details on the benchmark information.

**Baselines** We primarily compare our approach against models of comparable scale (∼7B parameters). The baselines include (i) closed-source models like GPT-4o [49], Claude for Computer Use [50], and Gemini 2.0 [51], as well as (ii) open-source models like SeeClick [34], ShowUI [39], and Magma [52]. We particularly highlight several baselines that share the same backbone as ours, including the backbone Qwen2-VL [38], Aguvis-7B [7], UGround-v1-7B [8], and UI-TARS-7B [10]. We also conduct performance comparison among Qwen-2.5-VL and models using it as backbone like Jedi [44]. Unless otherwise specified, all numbers are reported from the original paper or from the UI-TARS benchmark[10].

Table 3: Performance comparison on *ScreenSpot-v2*. [†] indicates results obtained from our own evaluation of the official model on Huggingface.

| | Mobile-Text | Mobile-Icon | Desktop-Text | Desktop-Icon | Web-Text | Web-Icon | Avg |
|---|---|---|---|---|---|---|---|
| Operator | 47.3 | 41.5 | 90.2 | 80.3 | 92.8 | 84.3 | 70.5 |
| GPT-4o + OmniParser-v2 | 95.5 | 74.6 | 92.3 | 60.9 | 88.0 | 59.6 | 80.7 |
| OS-Atlas-4B | 87.2 | 59.7 | 72.7 | 46.4 | 85.9 | 63.1 | 71.9 |
| UI-TARS-2B | 95.2 | 79.1 | 90.7 | 68.6 | 87.2 | 78.3 | 84.7 |
| **GUI-Actor-2B** | 95.0 | 82.2 | 92.2 | 81.8 | 92.9 | 82.7 | **88.6** |
| **GUI-Actor-2B + Verifier** | 95.9 | 81.5 | 94.3 | 82.9 | 93.6 | 82.8 | **89.3** |
| SeeClick-9.6B | 78.4 | 50.7 | 70.1 | 29.3 | 55.2 | 32.5 | 55.1 |
| Magma-8B | 62.8 | 53.4 | 80.0 | 57.9 | 67.5 | 47.3 | 61.5 |
| OS-Atlas-7B | 95.2 | 75.8 | 90.7 | 63.6 | 90.6 | 77.3 | 84.1 |
| Aguvis-7B[†] | 95.5 | 77.3 | 95.4 | 77.9 | 91.0 | 72.4 | 86.0 |
| UGround-V1-7B[†] | 95.0 | 83.3 | 95.0 | 77.8 | 92.1 | 77.2 | 87.6 |
| UI-TARS-7B | 96.9 | 89.1 | 95.4 | 85.0 | 93.6 | 85.2 | 91.6 |
| **GUI-Actor-7B** | 96.5 | 84.3 | 91.7 | 84.1 | 93.9 | 82.3 | **89.5** |
| **GUI-Actor-7B + Verifier** | 97.2 | 84.8 | 94.3 | 85.0 | 94.0 | 85.2 | **90.9** |
| UI-TARS-72B | 94.8 | 86.3 | 91.2 | 87.9 | 91.5 | 87.7 | 90.3 |

Table 4: Performance comparison of models based on the **Qwen-2.5-VL** backbone.

| *ScreenSpot-Pro:* | Dev | Creative | CAD | Scientific | Office | OS | Avg |
|---|---|---|---|---|---|---|---|
| Qwen2.5-VL-3B | 21.4 | 25.8 | 18.4 | 29.5 | 40.9 | 20.4 | 25.9 |
| Qwen2.5-VL-7B | 29.1 | 24.9 | 13.8 | 31.1 | 45.7 | 22.4 | 27.6 |
| Jedi-3B | 38.1 | 34.6 | 23.0 | 38.6 | 57.0 | 25.0 | 36.1 |
| Jedi-7B | 27.4 | 34.0 | 32.2 | 52.4 | 68.7 | 26.0 | 39.5 |
| **GUI-Actor-3B** | 39.8 | 36.7 | 34.1 | 49.6 | 61.3 | 35.2 | **42.2** |
| **GUI-Actor-3B + Verifier** | 43.8 | 37.8 | 43.3 | 48.4 | 63.5 | 42.9 | **45.9** |
| **GUI-Actor-7B** | 38.1 | 41.4 | 38.3 | 50.8 | 63.0 | 38.8 | **44.6** |
| **GUI-Actor-7B + Verifier** | 43.1 | 41.9 | 48.3 | 47.2 | 65.7 | 43.4 | **47.7** |

| *ScreenSpot-v2:* | Mobile-Text | Mobile-Icon | Desktop-Text | Desktop-Icon | Web-Text | Web-Icon | Avg |
|---|---|---|---|---|---|---|---|
| Qwen2.5-VL-3B | 93.4 | 73.5 | 88.1 | 58.6 | 88.0 | 71.4 | 80.9 |
| Qwen2.5-VL-7B | 97.6 | 87.2 | 90.2 | 74.2 | 93.2 | 81.3 | 88.8 |
| Jedi-3B | 96.6 | 81.5 | 96.9 | 78.6 | 88.5 | 83.7 | 88.6 |
| Jedi-7B | 96.9 | 87.2 | 95.9 | 87.9 | 94.4 | 84.2 | 91.7 |
| **GUI-Actor-3B** | 97.6 | 83.4 | 96.9 | 83.6 | 94.0 | 85.7 | **91.0** |
| **GUI-Actor-3B + Verifier** | 98.3 | 85.3 | 96.9 | 87.9 | 95.3 | 86.7 | **92.4** |
| **GUI-Actor-7B** | 97.6 | 88.2 | 96.9 | 85.7 | 93.2 | 86.7 | **92.1** |
| **GUI-Actor-7B + Verifier** | 96.9 | 89.6 | 97.4 | 86.4 | 95.7 | 84.7 | **92.5** |

**Main Results** Table 1, 2, 3, and 4 present performance comparisons on ScreenSpot-Pro, ScreenSpot, and ScreenSpot-v2 benchmarks. Our models, *GUI-Actor-2B* and *GUI-Actor-7B*, consistently outperform existing state-of-the-art methods across all benchmarks, with the 2B model even surpassing many competing 7B models. While there is one exception UI-TARS-7B achieving stronger performance, it benefits from training on a significantly larger dataset that includes both public and proprietary data (see Figure 1). Additionally, it undergoes a more extensive training pipeline, including continual pre-training, an annealing phase, and a final stage of direct preference optimization (DPO). Although our models are trained solely with supervised fine-tuning, they achieve competitive or even superior results on ScreenSpot-Pro, demonstrating its strong capability and potential.

**Robust Out-of-Distribution Generalization** As shown in Table 1, both *GUI-Actor-2B* and *GUI-Actor-7B* demonstrate strong performance on ScreenSpot-Pro—an out-of-distribution benchmark with higher resolutions and substantial domain shifts—surpassing the previous state-of-the-art UI-TARS model by +9.0 and +5.0 points with 2B and 7B parameters, respectively. We attribute this gain to explicit spatial-semantic alignment: unlike coordinate-based approaches such as UI-TARS, GUI-Actor leverages an attention-based action head that grounds semantic cues directly in discrete visual regions. This design embeds a stronger spatial inductive bias and naturally aligns with the patch-based representations of modern vision backbones. As a result, GUI-Actor is better equipped to reason over localized visual content, enabling robust generalization across diverse screen resolutions and UI layouts. Further evidence of this robustness is shown in the Figure 3(c): as training progresses, *GUI-Actor-2B* and *GUI-Actor-7B* show no sustained overfitting on the out-of-domain ScreenSpot-Pro benchmark: its accuracy recovers from early dips, then gradually increases before stabilizing. In contrast, baseline performance steadily declines after peaking early in training.

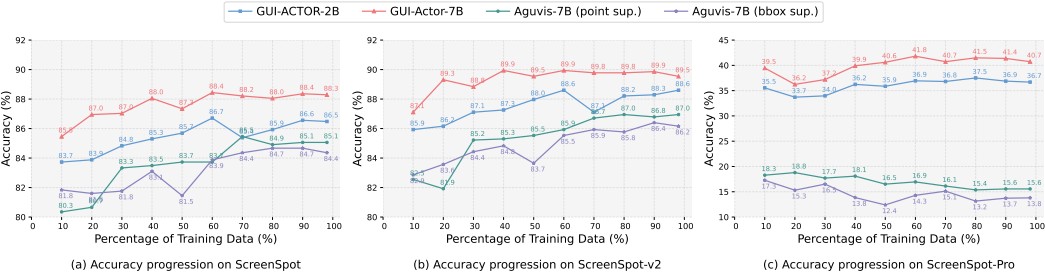

Figure 3: Accuracy Progression Over Training Steps.

**Improved Sample Efficiency** Figure 3 illustrates how `GUI-Actor`'s design leads to improved sample efficiency compared to coordinate-based baselines. `GUI-Actor` reaches its final accuracy on both ScreenSpot and ScreenSpot-v2 using only ∼60% of the training data, outperforming the point and box-supervised models of AGUVIS, which plateau after 80-90% of the data. This efficiency stems from `GUI-Actor`'s explicit spatial-semantic alignment through its action head, which enables grounding directly at the vision backbone's native patch resolution, avoiding the granularity mismatch that hampers baseline methods. Additionally, our multi-patch supervision strategy gracefully handles the supervision ambiguity in coordinate generation based methods, offering dense, spatially structured supervision signals.

**Enabling backbone VLM grounding on GUIs without sacrificing general-purpose strengths.**
We introduce a variant, *GUI-Actor-LiteTrain*, where we freeze all backbone VLM parameters and train only the newly introduced components for the action head and special tokens. This setup explores how `GUI-Actor` can impart GUI grounding capabilities without diminishing the VLM's general-purpose strengths. As shown in Table 5, *GUI-Actor-LiteTrain* yields substantial performance improvements over the unmodified backbone VLM. With the help of a grounding verifier, it even rivals fully fine-tuned coordinate generation models. These results suggest that the backbone VLM already exhibits strong perceptual understanding of UI screenshots. As such, training the model to generate coordinates in text format may primarily focus on coordinate mapping, offering limited contribution to the semantic understanding of UI elements. More importantly, *GUI-Actor-LiteTrain* retains the backbone's original language and vision-language capabilities, demonstrating that lightweight integration can enable grounding without compromising generality.

Table 5: Analysis on lightweight training (*-LiteTrain*), where the backbone Qwen-2-VL is frozen, and only the newly introduced parameters for the action head and special tokens are updated in training.

| Method | Updated # of Params | ScreenSpot-Pro | ScreenSpot | ScreenSpot-v2 |
|---|---|---|---|---|
| GUI-Actor-2B-LiteTrain | 19M | 25.4 | 71.4 | 73.9 |
| GUI-Actor-2B-LiteTrain + Verifier | 19M | 34.0 | 79.2 | 82.3 |
| GUI-Actor-7B-LiteTrain | 103M | 22.9 | 73.5 | 74.9 |
| GUI-Actor-7B-LiteTrain + Verifier | 103M | 35.8 | 81.3 | 83.8 |

**Boosting Performance via Grounding Verifier** The results in Tables 1, 2, 3, and 5 demonstrate that the grounding verifier consistently improves performance, highlighting its effectiveness in enhancing grounding accuracy. The improvement is especially significant on ScreenSpot-Pro, where it boosts GUI-Actor-7B by nearly 4 points and GUI-Actor-7B-LiteTrain by an impressive 13 points. Additionally, we investigate the benefits of the Verifier Self-Aggregation strategy in Appendix G.1 and evaluate the verifier's applicability to other baseline models in Appendix G.2. Our findings suggest that our verifier is highly robust and well-suited to GUI-Actor, as it requires only a single forward pass to generate diverse region proposals.

**Ablation Study** Table 6 present the results of our ablation study, where "bbox sup." and "point sup." denote models trained to predict the ground-truth bounding box or action point coordinates in natural language format, respectively. The results show that models trained with coordinate generation (both bounding box and point supervision) consistently underperform compared to *GUI-Actor-7B*

across the benchmarks, highlighting the effectiveness and necessity of explicit spatial-semantic alignment achieved through our proposed action head. Interestingly, despite having access to more spatial information, *Aguvis-7B (bbox sup.)* performs similarly to or worse than *Aguvis-7B (point sup.)*, suggesting that without architectural mechanisms or spatial inductive bias, these coordinate generation based methods remain disconnected from the underlying visual representation, limiting their generalization and grounding capabilities. For more detailed results on different domains and tasks, see Table 10 in the Appendix.

Table 6: Ablation study: averaged performance across benchmarks.

| Method | ScreenSpot-Pro | ScreenSpot | ScreenSpot-v2 |
|---|---|---|---|
| GUI-Actor-7B | 40.7 | 88.3 | 89.5 |
| Aguvis-7B (bbox sup.) | 13.8 | 84.4 | 84.4 |
| Aguvis-7B (point sup.) | 15.6 | 85.1 | 87.0 |

**Multi-Region Prediction Without Extra Inference Cost**   Due to its attention-based grounding mechanism, `GUI-Actor` can generate multiple candidate action regions in a single forward pass, incurring no extra inference cost. To evaluate the effectiveness of these high-probability regions, we use the Hit@k metric, where k represents the number of top-ranked predictions considered. Figure 4a shows that `GUI-Actor` exhibits a substantial improvement from Hit@1 to Hit@3, whereas the gap for baselines is relatively marginal. In our analysis, we observed that for coordinate-generation-based baselines, even when multiple predictions are sampled, the outputs are mostly identical, *e.g.*, shifting slightly from (0.898, 0.667) to (0.899, 0.666). In contrast, our model simultaneously produces multiple candidate regions from the attention distribution in a single forward pass. These candidates are mutually exclusive, naturally promoting diversity and increasing the chance of capturing all valid action regions. Figure 4b provides a qualitative example where our approach successfully identifies all ground-truth regions required for action execution.

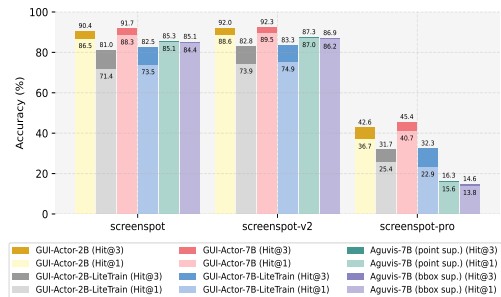

(a) Hit@1 and Hit@3 for different models.

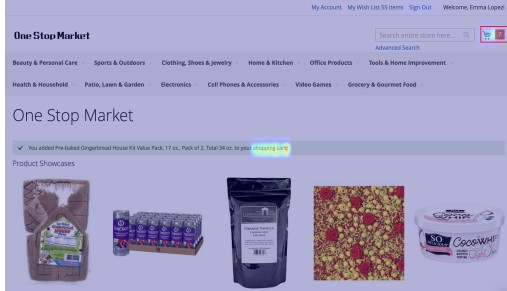

(b) GUI-Actor can capture multiple potential regions.

Figure 4: (a) Hit@1 and Hit@3 for different methods. For Aguvis baselines, we run inference 3 times with temperature = 1.0, top_p = 0.95. (b) Illustration of multi-region prediction. In this example, the instruction is "check shopping cart" and the central "shopping cart" text is clickable, while the ground truth is only the top-right icon.

## 6   Conclusion

We present `GUI-Actor`, a novel coordinate-free visual grounding framework for GUI agents. Departing from prevailing text-based coordinate generation paradigms, `GUI-Actor` introduces a dedicated `<ACTOR>` token that attends to target visual patches to directly localize GUI elements on the screen. This mechanism explicitly aligns spatial visual features with the semantic signals from instructions, and naturally supports bounding-box–based multi-patch supervision, which helps mitigate the ambiguity inherent in single-point predictions. Benefiting from its ability to propose multiple candidate regions in a single pass, `GUI-Actor` further employs a lightweight verifier to select the most plausible target at inference time. Experiments across diverse benchmarks demonstrate that `GUI-Actor` outperforms state-of-the-art methods and exhibits stronger generalization to unseen layouts and screen resolutions. We conduct extensive analyses to understand the effectiveness of each component in our framework, highlighting its promising potential for advancing visual GUI agents.

# Acknowledgments

We would like to thank Boyu Gou for providing the bounding box training data, and Yiheng Xu, Qiushi Sun, Zichen Ding, and Fangzhi Xu for their valuable discussions and insightful suggestions.

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

## A Limitations

Our attention-based action generation is particularly well-suited for GUI environments, where interface elements such as icons and text lines often exhibit regular geometric patterns that align natually with patch-based detection. However, a limitation stems from the base model we employ: the backbone VLM (*e.g.*, Qwen2-VL) adopts a Naive Dynamic Resolution strategy with a fixed patch size of $28 \times 28$ pixels. This poses challenges when dealing with very small interface elements (e.g., icons smaller than $10 \times 10$ pixels), as such fine-grained details may be insufficiently represented. Although this challenge is not unique to our method, it can be pronounced in tasks demanding high-precision control, such as those encountered in professional software like CAD tools. While we introduce a simple mitigation strategy in this work, fully addressing this limitation may require more substantial advancements in the future, such as improving the visual encoder's perceptual resolution or incorporating offset-based spatial refinement.

## B Details on Multi-Patch Supervision

A key advantage of our approach lies in its ability to leverage dense, spatially structured supervision from bounding boxes. Unlike traditional coordinate-based methods that depend on a single, potentially ambiguous click point, `GUI-Actor` enables supervision over entire target regions, capturing the spatial extent of actionable elements more effectively. An example is illustrated in Figure 2b.

Specifically, we convert each ground-truth bounding box into a binary mask over the $W \times H$ image patch grid. Given a normalized bounding box $b = [\text{left}, \text{top}, \text{right}, \text{bottom}] \in [0, 1]^4$, we scale the normalized coordinates to the patch grid resolution:

$$(\lfloor \text{left} \cdot W \rfloor, \ \lfloor \text{top} \cdot H \rfloor, \ \lceil \text{right} \cdot W \rceil, \ \lceil \text{bottom} \cdot H \rceil).\tag{8}$$

This rounding ensures that all patches partially or fully covered by the bounding box are included in the supervision region. All patches whose indices fall within the resulting grid-aligned region are labeled as positives (i.e., mask value 1), while all others are assigned a value of 0. This yields a binary vector $\mathbf{y} \in \{0, 1\}^M$ aligned with the image patch sequence, where $M = W \times H$.

We define the action head loss as the KL divergence between the predicted attention distribution $\{a_1, \ldots, a_M\}$ and a normalized target distribution $\mathbf{p}$ derived from the binary mask $\mathbf{y} \in \{0, 1\}^M$:

$$p_i = \frac{y_i}{\sum_{j=1}^{M} y_j + \epsilon}, \text{ for } i = 1, \ldots, M; \quad \mathcal{L}_{\texttt{<ACTOR>}} \ = \sum_{i=1}^{M} p_i \log \frac{p_i}{a_i},\tag{9}$$

where $\epsilon$ is a small constant for numerical stability.

## C Visualization of Attention Maps from GUI-Actor

To better understand the grounding behavior of GUI-Actor, we provide additional examples visualizing its attention maps overlaid on the original input images in Figure 5.

The following Python code outlines the visualization process: starting from the raw attention scores, we normalize and reshape them to match the image dimensions, apply a colormap for clearer interpretation, and finally blend the attention heatmap with the original image. This produces an intuitive overlay that highlights regions the model attends to when making decisions.

```python
width, height = image.size
W, H = attention_map_size # This may not equal width // patch_size due
    to image reshaping, and you may need W // 2 and H // 2 due to the
    Naive Dynamic Resolution operation in Qwen2-VL

scores = np.array(json_prediction['attn_scores'][0]).reshape(H, W)

# Normalize the attention weights for coherent visualization
scores_norm = (scores - scores.min()) / (scores.max() - scores.min())

# Resize the attention map to match the image size. You can choose
    different resize strategies, such as NEAREST and BILINEAR.
```

```
10  score_map = Image.fromarray((scores_norm * 255).astype(np.uint8)).
        resize((width, height), resample=Image.BILINEAR)
11
12  # Apply colormap
13  colormap = plt.get_cmap('jet')
14  colored_score_map = colormap(np.array(score_map) / 255.0)  # returns
        RGBA
15  colored_score_map = (colored_score_map[:, :, :3] * 255).astype(np.
        uint8)
16  colored_overlay = Image.fromarray(colored_score_map)
17
18  # Blend with original image
19  blended = Image.blend(image, colored_overlay, alpha=0.3)
```

Listing 1: Python code for overlaying the attention score map on the image.

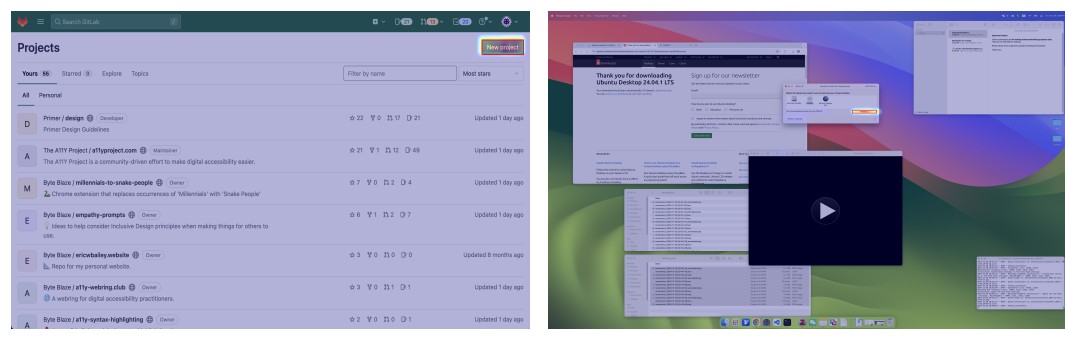

(a) ScreenSpot: "click the button to create a new project"          (b) ScreenSpot-Pro: "restart from CD"

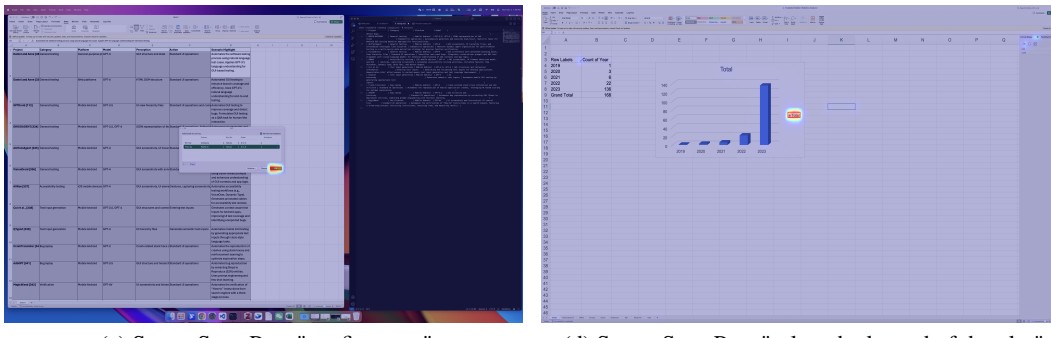

(c) ScreenSpot-Pro: "confirm sort"          (d) ScreenSpot-Pro: "select the legend of the plot"

Figure 5: Example visualizations from (a) ScreenSpot and (b)(c)(d) ScreenSpot-Pro. Each image shows the original interface with an overlaid attention map indicating regions of interest of GUI-Actor. The attention maps largely overlap with the ground truth areas (red bounding boxes), demonstrating that the model can effectively capture the accurate UI elements.

# D   Training Datasets used for GUI-Actor

We compile our training data from several publicly available, high-quality GUI datasets. Summary statistics are provided in Table 7. Note that we exclude samples from Wave-UI that overlap with downstream task test sets.

---
[1]https://huggingface.co/datasets/agentsea/wave-ui

Table 7: Overview of training datasets used for GUI-Actor.

| Dataset | # of Elements | # of Screenshots | Platform |
|---|---|---|---|
| Uground Web-Hybrid [8] | 8M | 775K | Web |
| GUI-Env [23] | 262K | 70K | Web |
| GUI-Act [23] | 42K | 13K | Web |
| AndroidControl [53] | 47K | 47K | Android |
| AMEX [24] | 1.2M | 100K | Android |
| Wave-UI[1] | 50K | 7K | Hybrid |
| **Total** | **9.6M** | **1M** | - |

## E GUI Visual grounding Benchmarks

In these benchmarks, each screenshot is paired with a natural language instruction written by human annotators, typically describing the content or function of the target element, e.g., "the new project button" or "switch to weekly view in the calendar." The agent is required to identify the location of the corresponding element on the screen based on the given instruction. **ScreenSpot** is the first benchmark specifically designed for GUI visual grounding, containing 1,272 single-step instructions paired with corresponding target elements. It covers a wide range of GUI platforms, including mobile (Android and iOS), desktop (macOS and Windows), and web environments, and categorizes elements into text-based or icon elements. **ScreenSpot-v2** is a corrected version of ScreenSpot that fixes annotation errors and ambiguous instructions, while keeping the total number of samples unchanged.

**ScreenSpot-Pro** is a recently introduced challenging benchmark tailored for high-resolution professional scenarios. It contains 1,581 tasks annotated by experts across 23 professional applications spanning three operating systems. Compared to ScreenSpot, ScreenSpot-Pro features higher-resolution screenshots and a larger domain gap from most grounding pretraining data, e.g., industrial software and multi-window interfaces. We view its performance as a practical estimate of generalization for GUI visual grounding models.

## F More Detailed on Grounding Verifier

### F.1 Data Construction

We construct the verifier training dataset from the OS-Atlas dataset [45], which spans desktop, mobile, and web domains. The original data consists of triplets in the form of (image, query, bounding box), where each image is paired with multiple queries and their corresponding bounding boxes. For each triplet, we generate a **positive example** by placing a marker at the center of the bounding box, treating it as the correct grounding point for the given query. To create **negative examples**, we apply two strategies: (1) selecting the center of a different bounding box from the same image to simulate a semantically plausible but incorrect location; (2) randomly sampling a point outside the correct bounding box to simulate an unrelated action. As shown in Figure 6, each proposed point is marked on the image with a hollow red circle. This process produces two labeled examples per query: one positive and one negative, formatted as:

$$\{\text{image with correct point}, \text{query}, \text{`True'}\}, \quad \{\text{image with wrong point}, \text{query}, \text{`False'}\}.$$

In total, we construct a balanced training set containing 730K examples, equally split between positive and negative cases. The overview of our dataset is listed in Table 8.

### F.2 Patch Selection

Given the top $M$ candidate patches from the attention map, our goal is to select the one that best aligns with the user instruction in the image. A straightforward approach is to draw a marker at the center of each patch and use a verifier to score how well each position satisfies the instruction $x$. Specifically, for each candidate image $I$ with a marked point, we use the verifier $\theta_v$ to compute the probability of predicting tokens 'True' or 'False': $P_{\text{true}} = P_{\theta_v}(\text{`True'}|I, x)$ and $P_{\text{false}} = P_{\theta_v}(\text{`False'}|I, x)$. We then define the score for each position as the normalized probability of the 'True' token: $\mathbf{s}(I, x) = \frac{P_{\text{true}}}{P_{\text{true}} + P_{\text{false}}}$.

Table 8: Overview of the dataset used to train our Grounding Verifier, including both positive and negative examples. **Since multiple positive and negative samples can be generated from a single screenshot, the size of our dataset can exceed that of the original dataset.**

| Dataset | # of Screenshots | Platform |
|---|---|---|
| SeeClick [34] | 147K | Web |
| FineWeb [54] | 123K | Web |
| UIbert [55] | 17K | Mobile |
| AMEX [24] | 155K | Android |
| RICOSCA [56] | 30K | Android |
| Widget Captioning [57] | 22K | Mobile |
| Linux-desktop [45] | 9K | Linux |
| Windows-desktop [45] | 220K | Windows |
| MacOS-desktop [45] | 7K | MacOS |
| **Total** | 730K | - |

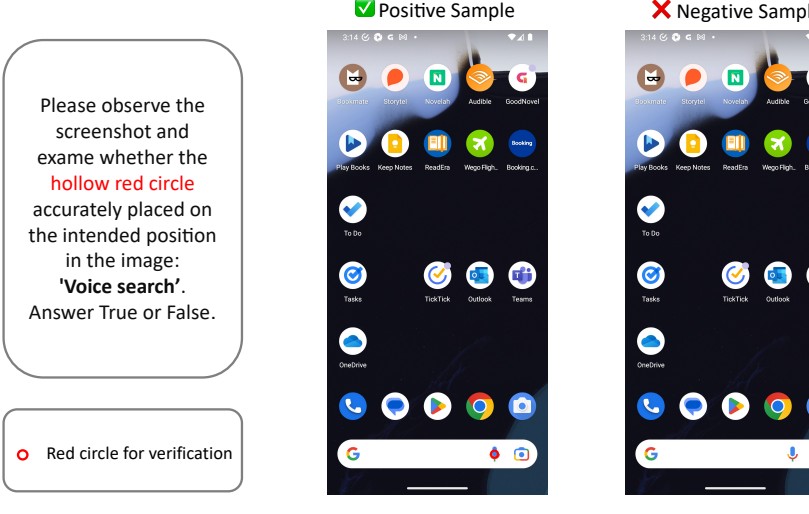

Figure 6: Illustration of positive and negative examples used to train the grounding verifier.

A key limitation of this approach is that each patch (typically $28 \times 28$ pixels) may miss small icons located between two neighboring patches, leading to inaccurate target localization. To address this, we introduce a simple yet effective refinement. We first cluster 4-connected neighboring patches and compute a weighted center based on the verifier scores of the individual patches. This enables the generation of candidate points that lie between adjacent patches and improves localization accuracy without directly modifying the patch size of the base model.

In our implementation, we use up to $M = 20$ top-scoring patches, filtering out those with attention weights below 20% of the maximum attention weight. We then apply clustering to the neighboring patches, compute the weighted centers of these clusters, and add them to the set of $M$ candidate positions. Each candidate position is scored using $\mathbf{s}(I, x)$, and we select the one with the highest score. Given a candidate coordinate $(x, y)$, we crop the image using a square region of size $l_{\text{crop}} \times l_{\text{crop}}$ centered at $(x, y)$. This is implemented as:

```
image.crop((
    max(0, x - l_crop//2),
    max(0, y - l_crop//2),
    min(image.size[0], x + l_crop//2),
    min(image.size[1], y + l_crop//2)
))
```

Table 9: Verifier Self-aggregation on ScreenSpot-Pro.

| | Dev | Creative | CAD | Scientific | Office | OS | Avg-Text | Avg-Icon | Avg |
|---|---|---|---|---|---|---|---|---|---|
| **GUI-Actor-7B + Verifier** | **40.1** | 39.0 | **37.2** | **47.2** | 63.5 | 41.8 | 61.1 | 16.7 | 44.2 |
| **GUI-Actor-7B + Verifier Self-aggregation** | 39.5 | **40.2** | **38.3** | 44.9 | **63.9** | **44.9** | **61.3** | **17.4** | **44.5** |

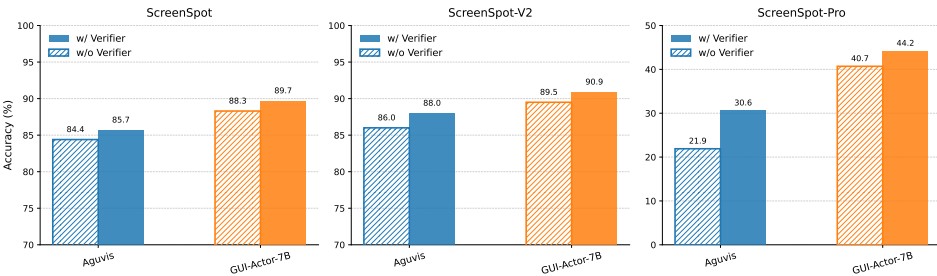

Figure 7: Comparison with AGUVIS using the verifier. AGUVIS inferences 21 times for verification. In contrast, GUI-Actor performs a single inference step, requiring only about 5% of the computation during inference.

We set $l_{crop} = 1000$ pixels for all tasks. To reduce the computational cost, if a candidate position achieves a high confidence score (e.g., $s(I, x) > 0.95$), we immediately return that position without evaluating the remaining candidates. In our experiments, we set the threshold to 0.95 for tasks in ScreenSpot-Pro and 0.8 for ScreenSpot and ScreenSpot-v2. A lower threshold reduces reliance on the verifier and instead trusts the grounding model's output, which is suitable when the grounding model is highly accurate. In contrast, a higher threshold prompts the verifier to more carefully assess each candidate, which is beneficial when the grounding model is less reliable, as in ScreenSpot-Pro.

# G    Improving Grounding with Verifier

## G.1    Enhancing Generation with Verifier Self-Aggregation

In this section, we explore how to further leverage the verifier's capability through a simple yet effective technique that we call Verifier Self-Aggregation (VS). The idea is to crop the input image at multiple scales and compute the verifier scores for each crop, then average these scores to obtain a more robust final prediction. This approach balances the trade-off between capturing detailed local information (with smaller crops) and maintaining a broader context (with larger crops). Specifically, we use two crop sizes in our experiments: $l_{crop} = 1200$ and $l_{crop} = 1400$ for ScreenSpot-pro. The results, shown in Table 9, demonstrate that verifier self-aggregation leads to improved performance on ScreenSpot-Pro. Verifier self-aggregation provides a simple yet effective strategy to enhance verifier robustness, while also highlighting the need for more robust verifiers in the future.

## G.2    Comparison with Baseline Using Verifier

To further validate the effectiveness of our grounding verifier, we integrate it into the AGUVIS baseline by sampling multiple candidate positions during inference and selecting the one with the highest verifier score. Specifically, AGUVIS generates one deterministic output at temperature 0.0 and samples 20 additional candidate points using a temperature of 0.7. While this approach explores a broader range of plausible locations, it incurs substantial computational overhead for these models.

In contrast, our GUI-Actor uses the attention map to propose multiple candidate points within a single pass. This leads to a much more efficient process—**requiring only about 5% of the computation** compared to AGUVIS—while achieving considerably higher grounding accuracy on ScreenSpot and ScreenSpot-v2, and significantly outperforming AGUVIS on the more challenging ScreenSpot-Pro benchmark. These results demonstrate both the efficiency of GUI-Actor and the general effectiveness of the verifier in improving action selection, even when applied to other models.

## H Detailed Numbers of Ablation Study

Table 10: Ablation study on *ScreenSpot-Pro*, *ScreenSpot*, and *ScreenSpot-v2*.

| ScreenSpot-Pro: | Dev | Creative | CAD | Scientific | Office | OS | Avg |
|---|---|---|---|---|---|---|---|
| GUI-Actor-7B | 38.8 | 40.2 | 29.5 | 44.5 | 56.5 | 36.2 | 40.7 |
| Aguvis-7B (bbox sup.) | 12.4 | 17.0 | 1.5 | 18.1 | 21.7 | 11.7 | 13.8 |
| Aguvis-7B (point sup.) | 15.7 | 19.4 | 3.8 | 17.3 | 24.4 | 11.7 | 15.6 |
| ScreenSpot: | Mobile-Text | Mobile-Icon | Desktop-Text | Desktop-Icon | Web-Text | Web-Icon | Avg |
| GUI-Actor-7B | 94.9 | 82.1 | 91.8 | 80.0 | 91.3 | 85.4 | 88.3 |
| Aguvis-7B (bbox sup.) | 92.3 | 73.4 | 92.3 | 76.4 | 92.6 | 74.8 | 84.4 |
| Aguvis-7B (point sup.) | 92.3 | 79.0 | 93.3 | 71.4 | 92.6 | 75.2 | 85.1 |
| ScreenSpot-v2: | Mobile-Text | Mobile-Icon | Desktop-Text | Desktop-Icon | Web-Text | Web-Icon | Avg |
| GUI-Actor-7B | 96.5 | 84.3 | 91.7 | 84.1 | 93.9 | 82.3 | 89.5 |
| Aguvis-7B (bbox sup.) | 92.3 | 73.4 | 92.3 | 76.4 | 92.6 | 74.8 | 84.4 |
| Aguvis-7B (point sup.) | 96.1 | 80.1 | 96.1 | 74.6 | 93.6 | 74.3 | 87.0 |

## I Online Benchmark Evaluation on OS-World-W

To evaluate the real-world effectiveness of our proposed `GUI-Actor`, we conduct experiments on OS-World [16] using *GUI-Actor-7B* for quick validation. OS-World is a live benchmark designed to test GUI agents in realistic desktop environments. We focus on a curated subset of 49 Windows-specific tasks, denoted as OSWorld-W, covering a variety of multi-step office and multi-application scenarios. Each task is paired with handcrafted verification scripts to ensure reliable automatic evaluation.

Following the standard evaluation pipeline, we adopt GPT-4o as the planner. At each step, the planner observes the current GUI screenshot and user instruction and generates a natural language plan. This plan is then grounded into concrete actions—either via coordinate-based or coordinate-free mechanisms—by the underlying grounding model, which plays a critical role in determining the agent's success.

We compare `GUI-Actor` with several leading visual grounding baselines: Aguvis-7B [7], NAVI [58], and OmniAgent [59]. As shown in Table 11, `GUI-Actor` achieves the highest task success rate at 12.2%, outperforming OmniAgent and NAVI (both at 10.2%) and substantially surpassing Aguvis-7B (4.0%). These results highlight the effectiveness and robustness of `GUI-Actor` in complex, real-world GUI environments. Despite having no exposure to OSWorld-W tasks during training, `GUI-Actor` generalizes well to unseen scenarios, delivering more accurate and reliable grounding performance than existing alternatives.

Table 11: Task Success Rate on the OSWorld-W subset (49 live Windows GUI tasks). `GUI-Actor` significantly outperforms existing grounding models in this real-world setting.

| Grounding Model | Success Rate (%) | #Tasks Completed |
|---|---|---|
| Aguvis-7B (point sup.) | 4.00 | 2/49 |
| NAVI | 10.2 | 5/49 |
| OmniAgent | 10.2 | 5/49 |
| GUI-Actor-7B (Ours) | **12.2** | **6/49** |

## J Grounding Small UI Elements

### J.1 Quantitative Analysis

A key limitation of GUI-Actor lies in the dependency on the backbone vision-language model's fixed patch size, which makes it challenging to localize very small UI elements that may occupy less than one visual patch. This limitation can lead to errors when such elements fall entirely within a single patch, preventing accurate grounding.

To quantify this issue, we analyze performance on the ScreenSpot-Pro[48] dataset, where targets are grouped by bounding box area relative to the patch size ($n = 14 \times 14$ pixels). As summarized in

Table 12, grounding accuracy for the smallest UI elements $[0, n)$ is significantly lower than for larger ones, highlighting the difficulty of fine-grained localization. Nevertheless, our attention-weighted refinement strategy—which computes the weighted center of top-$K$ patches—consistently improves accuracy across all size ranges, including the most challenging small-target cases.

Table 12: Accuracy by target UI element size on ScreenSpot-Pro, measured by area, $n = 14 \times 14$, without Verifier.

| Range | # of Examples | Top-1 Patch Center | Weighted Sum of Patch Cluster |
|---|---|---|---|
| $[0, n)$ | 23 | 4.35% | 13.04% ($\uparrow$ 200%) |
| $[n, 4n)$ | 349 | 6.59% | 11.46% ($\uparrow$ 73.91%) |
| $[4n, 9n)$ | 367 | 14.99% | 25.07% ($\uparrow$ 67.27%) |
| $[9n, )$ | 842 | 42.04% | 55.58% ($\uparrow$ 32.20%) |
| **All** | **1581** | **27.39%** | **38.14% ($\uparrow$ 39.26%)** |

## J.2 "Zoom-In" Strategy

To further alleviate this limitation, here we study a "Zoom-In" strategy, in which GUI-Actor first performs a coarse prediction and then zooms in by a factor of $2\times$ around the predicted region for a second round of grounding. This hierarchical approach allows finer spatial reasoning and substantially improves robustness in high-resolution scenarios. The results in Table 13 demonstrate the effectiveness of this strategy, showing notable gains in overall performance on ScreenSpot-Pro.

These analyses collectively demonstrate that small UI elements, though rare in typical desktop environments, pose meaningful challenges for patch-based grounding models. Our proposed refinement and zoom-in mechanisms effectively mitigate these errors, while future work may further explore multi-scale feature integration or offset-based localization to enhance fine-grained grounding.

Table 13: Effectiveness of the "Zoom-In" strategy on ScreenSpot-Pro.

| Model for Zoom-In Round | Model for Final Round | Zoom-In Ratio | ScreenSpot-Pro |
|---|---|---|---|
| / | Aguvis-7B | / | 22.9 |
| / | GUI-Actor-7B (w/o Verifier) | / | 38.1 |
| GUI-Actor-7B (w/o Verifier) | Aguvis-7B | $2\times$ | 43.0 |
| GUI-Actor-7B (w/o Verifier) | GUI-Actor-7B (w/o Verifier) | $2\times$ | **50.3** |

## K  Error Analysis on GUI Grounding Failures

To better understand the causes of grounding failures, we conducted a detailed error analysis across 40 examples from ScreenSpot-Pro and ScreenSpot-v2, respectively. The breakdown in Table 14 categorizes common failure types and highlights the influence of dataset characteristics such as instruction ambiguity, non-unique valid regions, and the size of target UI elements.

Table 14: Error analysis on 40 examples from ScreenSpot-Pro and ScreenSpot-v2, respectively.

| Error Type | ScreenSpot-Pro | ScreenSpot-v2 |
|---|---|---|
| The ground-truth valid action region is not unique (model captures another valid region not labeled as ground truth). | 7.50% | 7.50% |
| The instruction is ambiguous (model predicts a reasonable but non-annotated action). | 7.50% | 20.00% |
| The ground-truth region is smaller than 14px $\times$ 14px. | 5.00% | 0.00% |
| The ground-truth region gains some attention but fails to produce an accurate final position. | 47.50% | 27.50% |
| The ground-truth region gains little or no attention. | 32.50% | 45.00% |

This analysis reveals several key observations. First, many grounding errors stem from ambiguous or under-specified instructions, where multiple reasonable actions exist but only one is annotated as ground truth. Second, small UI elements ($< 14 \times 14$ px) continue to present challenges for fine-grained localization, even with our weighted refinement strategies. Finally, a substantial portion of errors arises when the model attends to the correct region but fails to precisely generate the final click coordinate.

We believe these findings provide valuable guidance for future dataset design and model refinement, especially regarding annotation granularity and instruction disambiguation.

## L    Ablation on Multi-Patch Supervision and Threshold Sensitivity

The multi-patch supervision during training is determined objectively. As described in Section 3 and illustrated in Figure 2b, our multi-patch supervision strategy labels all image patches that are partially or fully covered by the ground-truth bounding box as positive (label = 1), and all other patches as negative (label = 0). This binary supervision is applied purely based on spatial overlap, without any threshold tuning.

Regarding the threshold used at inference time to aggregate multiple clustered patches into a final prediction, we conducted an ablation study by varying the value from 0.1 to 1.0. As shown in Table 15, performance remains stable within a moderate range. However, extreme values—either too permissive (0.1) or too restrictive (1.0, equivalent to using only the top-1 patch)—lead to noticeable drops in accuracy. These results suggest that while the threshold choice does have an effect on performance, the method remains robust across a broad range of reasonable settings. This demonstrates that the attention-based aggregation is resilient to moderate variations in patch selection criteria.

Table 15: Ablation on the threshold used to produce the final click point from the attention map.

|  | ScreenSpot-v2 | ScreenSpot-Pro |
| --- | --- | --- |
| 0.1 | 91.4 | 39.0 |
| 0.2 | 91.7 | 42.7 |
| 0.3 (Reported in Table 4) | 92.1 | 44.6 |
| 0.35 | 92.2 | 45.3 |
| **0.4** | **92.2** | **46.4** |
| 0.45 | 92.0 | 45.9 |
| 0.5 | 91.7 | 46.0 |
| 0.6 | 90.6 | 45.5 |
| 0.7 | 90.3 | 43.5 |
| 0.8 | 89.3 | 41.4 |
| 0.9 | 87.1 | 39.7 |
| 1.0 (Top-1 Patch) | 85.1 | 36.9 |

