# OpenReview forum: "GUI-Actor: Coordinate-Free Visual Grounding for GUI Agents"
_NeurIPS.cc/2025/Conference — NeurIPS 2025 poster_

### Official Review · Reviewer_YmmE · 2025-06-23

**Clarity:** 4
**Significance:** 3
**Originality:** 3
**Rating:** 5
**Confidence:** 4

**Summary:**

This paper introduces GUI-Actor, a vision-language model for GUI grounding tasks that replaces text-based coordinate prediction with an attention-driven, “coordinate-free” grounding mechanism. The authors align a special <ACTOR> token directly with relevant visual patches through an action-head (MLPs), then apply a lightweight verifier to filter the top-K attended regions to get the final action point. Trained on ∼1 M screenshots, GUI-Actor achieves good element-accuracy on ScreenSpot, ScreenSpot-v2, and the challenging ScreenSpot-Pro benchmarks even if the only trained model parameters are the 100M-parameter action-head.

**Questions:**

1. The proposed method sets a confidence threshold, but it is not clear how the threshold is defined, and what if no candidate is above the threshold.
2. The author demonstrated fine-tuning the Qwen-2-VL model only. Is it possible to apply the proposed coordinate-free training to a existing GUI model such as UI-TARS-7B?

**Ethical Concerns:**

["NO or VERY MINOR ethics concerns only"]

**Final Justification:**

I maintain my positive score and recommend acceptance. The authors addressed all major concerns thoroughly:

- Comparison fairness: Ablations show GUI-Actor outperforms baselines even without the verifier.
- Efficiency: The verifier is lightweight, with early stopping and vLLM inference reducing latency. The design is reasonable for deployment.
- Small target precision: The proposed “Zoom-In” strategy effectively handles fine-grained elements.
- Generality: GUI-Actor works well with other VLMs like UI-TARS, showing broad applicability.
- Conceptual clarity: Follow-up clarified connections to coordinate-free grounding in prior work.

**Limitations:**

yes

**Quality:**

3

**Strengths And Weaknesses:**

Strengths: The work is methodologically novel: it drops brittle numeric-coordinate generation in favor of an explicit spatial–semantic attention map, naturally allowing dense multi-patch supervision and diverse candidate proposals. The narrative is clear, well-situated in recent GUI-agent literature, and backed by thorough ablations.

Weaknesses:
1. Comparing GUI-Actor-7B + verifier to baselines that predict actions in a single pass overstates its advantage, because the verifier injects an extra round of visual evidence and decision logic that competing methods never see. A like-for-like comparison should disable the verifier or add an equivalent post-filter to the baselines.
2. Running the GUI-Actor-7B + verifier increases GPU memory and adds an extra forward pass per click, introducing latency that is hard to hide in real-time GUI agents.
3. The proposed method makes the “coordinate-free” pointer to land only at patch centers. The 28 × 28 patch grid becomes too large and coarse when the input resolution is small, making fine-grained targets (e.g., tiny icons or narrow scroll bars) unreachable.

---

> ### Author Rebuttal · Authors · 2025-07-31
>
> Thank you for reviewing our work and providing valuable feedback. We have carefully addressed your concerns below. Please let us know if you have any further questions.
>
> **W1:** _Comparing GUI-Actor-7B + verifier to baselines that predict actions in a single pass overstates its advantage, because the verifier injects an extra round of visual evidence and decision logic that competing methods never see. A like-for-like comparison should disable the verifier or add an equivalent post-filter to the baselines._
>
> Thank you for the valuable comment. We reported the performance of GUI-Actor-7B both **with** and **without** the Verifier in our ablation studies (see Tables 4 and 5). To facilitate a more direct comparison, we reorganize key results and extend the evaluation to Qwen2.5-VL based models, as shown in **_Table H_** and **_Table I_** below. We can see that **_even without the Verifier, GUI-Actor consistently outperforms all SFT counterparts_** (OS-Atlas, Aguvis, and UGround). Notably, on the challenging OoD benchmark ScreenSpot-Pro, it also surpasses UI-TARS (trained on a substantially larger dataset and using a more complex 3-stage pipeline). This underscores the inherent strength of the GUI-Actor grounding model and its robustness to distribution shift. We will include this comparative table in the next revision.
>
> **_Table H._** _Performance Comparison across benchmarks for various models using Qwen2-VL as backbone._
> | Method                    | Backbone VLM | ScreenSpot-Pro | ScreenSpot | ScreenSpot-v2 |
> |---------------------------|---------------|----------------|-------------|----------------|
> | OS-Atlas-7B               | Qwen2-VL      | 18.9           | 82.5        | 84.1           |
> | AGUVIS-7B                 | Qwen2-VL      | 22.9           | 84.4        | 86.0          |
> | UGround-V1-7B             | Qwen2-VL      | 31.1           | 86.3        | 87.6          |
> | UI-TARS-7B                | Qwen2-VL      | 35.7           | **89.5**    | **91.6**       |
> | GUI-Actor-7B w/o Verifie             | Qwen2-VL      | **38.1**       | 86.6        | 88.9           |
> | GUI-Actor-7B w/ Verifier   | Qwen2-VL      | **41.6**       | **87.8**        | **89.8**           |
>
> **_Table I._** _Performance Comparison across benchmarks for various models using Qwen2.5-VL as backbone._
> | Method                     | Backbone VLM  | ScreenSpot-Pro | ScreenSpot-v2 |
> |----------------------------|----------------|----------------|----------------|
> | **_7B models:_**             |                |                |                |
> | Qwen2.5-VL-7B              | Qwen2.5-VL     | 27.6           | 88.8           |
> | Jedi-7B                   | Qwen2.5-VL     | 39.5           | 91.7           |
> | GUI-Actor-7B  w/o Verifie             | Qwen2.5-VL     | **44.6**       | **92.1**       |
> | GUI-Actor-7B w/ Verifier   | Qwen2.5-VL     | **47.7**           | **92.5**           |
> | **_3B models:_**             |                |                |                |
> | Qwen2.5-VL-3B              | Qwen2.5-VL     | 25.9           | 80.9           |
> | Jedi-3B                   | Qwen2.5-VL     | 36.1           | 88.6           |
> | GUI-Actor-3B  w/o Verifie              | Qwen2.5-VL     | **42.2**       | **91.0**       |
> | GUI-Actor-3B w/ Verifier   | Qwen2.5-VL     | **45.9**           | **92.4**           |
>
>
> **W2:** _Running the GUI-Actor-7B + verifier increases GPU memory and adds an extra forward pass per click, introducing latency that is hard to hide in real-time GUI agents._
>
> Thank you for the insightful comment. We agree that adding the verifier introduces additional computational overhead. Our **_motivation_** for integrating the verifier is to **_explore test-time scaling_** for GUI-Actor. Unlike typical GUI grounding models, **_GUI-Actor can naturally propose multiple diverse candidate regions in a single pass, making it well-suited for pairing with a verifier at test time._**
>
> To **_reduce inference time_**, we **_adopt three strategies_**: (1) using a lightweight 2B verifier, (2) applying early stopping when the verifier’s confidence score exceeds 0.95, effectively reducing the average number of verifier calls to 3.3 per sample on ScreenSpot-v2, and (3) leveraging vLLM for efficient inference.
>
> Considering real-time GUI agents, we have some findings below to support this design. For example, **_GUI-Actor-2B combined with a 2B verifier outperforms standalone 7B grounding models on the ScreenSpot-Pro benchmark_**. This means that with proper parallelization, the overall runtime is approximately equivalent to two 2B model inferences—still **_notably more efficient than running a 7B model—while achieving superior performance_**. This makes our approach particularly **_promising for deployment in real-time GUI agents on resource-constrained devices_** such as phones, which may not support 7B models.
>
> **_Table J._** _Performance comparison across different model configurations._
> | Base Model                   | GUI-Actor-2B w/o Verifier | GUI-Actor-2B w/ Verifier | GUI-Actor-7B w/o Verifier | UI-TARS-7B |
> |-----------------------------|---------------------------|---------------------------|----------------------------|------------|
> | Performance on ScreenSpot-Pro | 36.7                      | 41.8                      | 40.7                       | 35.7       |
>
>
> **W3:** _The proposed method makes the “coordinate-free” pointer to land only at patch centers. The 28 × 28 patch grid becomes too large and coarse when the input resolution is small, making fine-grained targets (e.g., tiny icons or narrow scroll bars) unreachable._
>
> Thank you for the thoughtful feedback. Theoretically, GUI-Actor is capable of locating elements smaller than 28x28 but larger than 14x14. The challenge of small element localization does exist for those **_smaller than 14x14_**, e.g., when they are located at the corner patch of a 28x28 grid. Fortunately, these extreme cases are **_relatively rare_** in typical desktop scenarios, as shown in **_Table K_**.
>
> To **_mitigate this limitation_**, we explored a simple yet effective **_"Zoom-In" strategy_** for GUI-Actor, where GUI-Actor makes an initial coarse prediction, then zooms in 2× around the predicted location and performs a second round of grounding. **_Table L_** shows this significantly boosts performance on ScreenSpot-Pro, demonstrating its value in high-resolution settings with small UI elements.
>
> **_Table K._** _Frequency of small UI elements (longest edge < 14 px) fully contained within a single patch._
> | Dataset         | # of Examples | # of with Small Elements | # of Small Element In One Patch |
> |-----------------|----------------------|--------------| --------------|
> | ScreenSpot-v2    | 1272              | 0                    | 0            |
> | ScreenSpot-pro   | 1581              | 52                   | 10            |
>
> **_Table L._** _Effectiveness of the "Zoom-In" strategy on ScreenSpot-Pro._
> | Model for Zoom-In Round              | Model for Final Round              | Zoom-in Ratio | Screenspot-Pro |
> |-------------------------------------|------------------------------------|----------------|----------------|
> | /                                   | Aguvis-7B                          | /              | 22.9           |
> | /                                   | GUI-Actor-7B                       | /              | 38.1           |
> | GUI-Actor-7B (w/o Verifier)         | Aguvis-7B                          | 2x             | 43.0           |
> | GUI-Actor-7B (w/o Verifier)         | GUI-Actor-7B (w/o Verifier)        | 2x             | 50.3           |
>
>
> **Q1:** _The proposed method sets a confidence threshold, but it is not clear how the threshold is defined, and what if no candidate is above the threshold._
>
> Thank you for pointing out the confusion. The verifier evaluates candidates sequentially, and we use an **_intuitive confidence threshold for early stopping_**: the first candidate exceeding this threshold is selected as the final prediction. **_If no candidate meets the threshold, we fall back to the one with the highest verifier score_**. We will clarify this in the revision. This strategy effectively reduces the average number of verifier calls from 20 to just 3.3 per sample on ScreenSpot-v2, with only a negligible impact on performance.
>
>
> **Q2:** _The author demonstrated fine-tuning the Qwen-2-VL model only. Is it possible to apply the proposed coordinate-free training to an existing GUI model such as UI-TARS-7B?_
>
> Thank you for the insightful question. Here we further apply our coordinate-free GUI-Actor framework to UI-TARS-7B. As shown in **_Table M_**, when using UI-TARS-7B as the base model, GUI-Actor achieves **_comparable performance on the ScreenSpot-v2_** benchmark and **_higher accuracy on the more challenging ScreenSpot-Pro benchmark_**. We attribute this improvement to UI-TARS-7B’s strong visual encoder, which is initialized from Qwen2VL and further trained on GUI-specific datasets, likely leading to richer and more semantically aligned representations. When combined with our action-head-based grounding approach, these enhanced features enable more precise localization. These findings suggest that **_our method extends beyond general-purpose VLMs and can effectively complement and enhance existing GUI models_**.
>
> **_Table M._** _Evaluation of GUI-Actor applied to two different base VLMs: Qwen2VL-7B and UI-TARS-7B-SFT._
> | Model                                                      | Screenspot-v2 | Screenspot-Pro |
> |------------------------------------------------------------|----------------|-----------------|
> | UI-TARS-7B                                                 | 91.6           | 35.7            |
> | GUI-Actor w/ Qwen2VL-7B (w/o Verifier)                     | 88.9           | 38.1            |
> | GUI-Actor w/ UI-TARS-7B (w/o Verifier)                     | 91.4           | 40.5            |

---

> > ### Comment · Reviewer_YmmE · 2025-08-05
> >
> > Thank you for your responses, which helps a lot for a better understanding.
> >
> > I am impressed by the effectiveness of the proposed action-head-based method. It makes me further wondering if this method is conceptually connected to any other coordinate-free method in computer vision domain? If there are some other related works on coordinate-free grounding, could you share some insights?

---

> > > ### Author Response · Authors · 2025-08-06
> > > **Thank you for your response!**
> > >
> > > We thank the reviewer for the thoughtful follow-up and are glad our previous responses helped improve understanding.
> > >
> > > Conceptually, our GUI-Actor can be viewed as a vision analogue of the **Pointer Networks** [1], originally proposed for solving geometric problems such as computing planar convex hulls, Delaunay triangulations, and the Travelling Salesman Problem, and later extended to NLP tasks like summarization [2]. The core idea is to **_learn a conditional probability distribution over input elements_**. In our case, the `<ACTOR>` token plays this role, attending over discrete visual tokens (patches) and enabling selection within the input set, rather than predicting continuous coordinates.
> > >
> > > Regarding related **_coordinate-free practices in computer vision_**, a relevant instantiation can be found in pixel-level segmentation works such as **_LISA_** [3]. LISA **_expands the original VLM vocabulary with a new token_**, whose last-layer embedding acts as a query for segmentation. At the same time, it employs an **_external vision encoder_** (e.g., SAM's image encoder) to obtain **dense pixel-level features**. These features, along with the new-token embedding, are passed to a **_dedicated segmentation decoder_** (e.g., SAM's mask decoder) to produce the final segmentation mask.
> > >
> > > While LISA shares the same high-level spirit with GUI-Actor—avoiding explicit (x, y) regression by predicting a distribution over dense visual units (pixels) using the VLM's last-layer representation of an anchor token—they **_differ significantly in where and how the distribution is formed_**. GUI-Actor eliminates the need for an additional vision encoder or decoder, and instead leverages the native vision–language alignment in modern VLMs. Specifically, the last-layer representation of the `<ACTOR>` token directly attends to and activates the relevant patch-level features within the backbone VLM itself using similarity and softmax, yielding a lightweight, decoder-free grounding mechanism.
> > >
> > > We thank the reviewer for the insightful comment and will reflect this in the final version.
> > >
> > > ---
> > >
> > > [1]  Pointer networks. NeurIPS 2015.
> > >
> > > [2] Get To The Point: Summarization with Pointer-Generator Networks. ACL 2017.
> > >
> > > [3] LISA: Reasoning Segmentation via Large Language Model. CVPR 2024.

---

> > > > ### Comment · Reviewer_YmmE · 2025-08-07
> > > >
> > > > Thanks for the response. I have no more questions. I will keep my positive score for the paper.

---

> > > > > ### Author Response · Authors · 2025-08-09
> > > > > **Thank you!**
> > > > >
> > > > > Thank you for your time and effort in reviewing our work!

---

### Official Review · Reviewer_8yA9 · 2025-06-29

**Clarity:** 3
**Significance:** 3
**Originality:** 3
**Rating:** 4
**Confidence:** 5

**Summary:**

This paper mainly focuses on the GUI-grounding problem in the GUI agent topic. The authors propose an interesting attention-based action head for the candidate action elements prediction and a grounding verifier to select the most plausible action region as the final grounding prediction. Several experimental results have been presented to show the effectiveness of the proposed method.

**Questions:**

Please refer to the weaknesses mentioned above.

**Ethical Concerns:**

["NO or VERY MINOR ethics concerns only"]

**Final Justification:**

The authors have addressed my concerns, and I will maintain my positive rating.

**Limitations:**

yes

**Quality:**

3

**Strengths And Weaknesses:**

Strengths:
1. The presentation and writing of the paper are clear and well organized.
2. Exploring attention-based grounding—rather than directly predicting coordinates via text—is both interesting and conceptually sound.
3. Introducing an < ACTOR > token as a grounding anchor for computing attention with image patch embeddings is a good design choice.

Weaknesses:
1. The action head requires the model to output the < ACTOR > token. For GUI-Actor-7B with LiteTrain, the VLM parameters remain frozen—how is the model reliably induced to produce this new < ACTOR > token, and was this token newly added to the vocabulary?

2. Tables 4 and 5 show that GUI-Actor-7B (no Verifier) outperforms GUI-Actor-7B w/ LiteTrain (no Verifier) by a substantial margin. Could you analyze which component drives this gain? Is it better < ACTOR > formatting, more accurate action-head predictions, or something else?

3. Adding the Verifier (Tables 4 and 5) yields a 3–9% boost in performance. Why does the action head’s direct attention map exhibit such a significant bias? Although the action model (7B) is much larger than the Verifier (2B), the action head’s direct attention maps still show substantial bias. One would expect the stronger model to leverage its internal reasoning capacity without external checks—could this indicate that the action head was not trained well enough to fully elicit the VLM’s inherent reasoning knowledge?

4. In the setup on L215, the Verifier may be invoked up to K = 20 times, introducing extra runtime and compute cost. Please include an inference-cost comparison against baseline models and report the average number of Verifier calls per sample.

Minor Questions:
1. Is the self-attention applied to image-patch embeddings in the attention-based action head causal or non-causal? Could this choice affect performance?

2. During GUI-Actor-7B’s supervised fine-tuning, which VLM parameters were actually trained?

---

> ### Author Rebuttal · Authors · 2025-07-31
>
> Thank you for reviewing our work and providing valuable feedback. We have carefully addressed your concerns below. Please let us know if you have any further questions.
>
> **W1:** _The action head requires the model to output the < ACTOR > token. For GUI-Actor-7B with LiteTrain, the VLM parameters remain frozen—how is the model reliably induced to produce this new < ACTOR > token, and was this token newly added to the vocabulary?_
>
>  Yes, the `<ACTOR>` token is newly added to the vocabulary. As noted, in the GUI-Actor-7B w/ LiteTrain setting, all original VLM parameters are frozen; only the `<ACTOR>` token embedding and the parameters introduced in the Action Head are trainable.
>
> During inference, if the model is **_only performing the GUI grounding_** task, it is **_not necessary for it to generate the `<ACTOR>` token_**. Instead, in the **_LiteTrain_** setting, we **_directly include the `<ACTOR>`token in the input prompt_**, e.g., "pyautogui.click(`<ACTOR>`)", and **_extract its final-layer hidden representation_**. This representation is then passed to the action head for grounding.
>
>
> **W2:** _Tables 4 and 5 show that GUI-Actor-7B (no Verifier) outperforms GUI-Actor-7B w/ LiteTrain (no Verifier) by a substantial margin. Could you analyze which component drives this gain? Is it better `<ACTOR>` formatting, more accurate action-head predictions, or something else?_
>
> **_The `<ACTOR>` token formatting is the same in both settings_**, so the performance gap is not due to differences in prompt structure.
> We believe the **_performance gain_** of GUI-Actor-7B (without Verifier) over GUI-Actor-7B with LiteTrain (also without Verifier) **_mainly comes from the difference in how many parameters of the model are fine-tuned_**. In the full fine-tuning setting, the entire model—including the ViT backbone—is updated, allowing for deeper task-specific adaptation. In contrast, the LiteTrain variant freezes the backbone and only updates the `<ACTOR>` token embedding and the Action Head, which limits its ability to fully align multimodal representations for the grounding task. In summary, while **_LiteTrain offers a more efficient training approach, full fine-tuning leads to stronger task adaptation and better overall performance_**.
>
>
> **W3:** _Adding the Verifier (Tables 4 and 5) yields a 3–9% boost in performance. Why does the action head’s direct attention map exhibit such a significant bias? Although the action model (7B) is much larger than the Verifier (2B), the action head’s direct attention maps still show substantial bias. One would expect the stronger model to leverage its internal reasoning capacity without external checks—could this indicate that the action head was not trained well enough to fully elicit the VLM’s inherent reasoning knowledge?_
>
> Thanks for the insightful comments. While it's true that adding a smaller Verifier improves performance, we believe it's **_not sufficient to conclude that the action head was inadequately trained or unable to leverage the VLM’s reasoning capacity_**.
>
> The Verifier is trained as a binary classifier to assess whether a given candidate position is plausible, whereas the grounding model learns to predict a full attention map over all input image patches. These two tasks, i.e., evaluation versus direct prediction, differ fundamentally in complexity. As noted in Line 176, **_selecting the correct option from a list (as the Verifier does) is generally easier than directly generating a precise location from an unconstrained space_**.
>
> Importantly, the performance boost from the Verifier suggests that the grounding model often identifies the correct region but may not consistently assign it the highest score, especially in cases involving ambiguity or multiple plausible targets. In these situations, the Verifier helps resolve uncertainty by selecting the most semantically appropriate candidate. **_Rather than compensating for a weak grounding model, the Verifier serves as a complementary module that enhances decision-making in challenging scenarios_**. For instance, we observe greater improvement on the more difficult benchmark, ScreenSpot-Pro.
>
>
> **W4:** _In the setup on L215, the Verifier may be invoked up to K = 20 times, introducing extra runtime and compute cost. Please include an inference-cost comparison against baseline models and report the average number of Verifier calls per sample._
>
> Thank you for your constructive suggestion. We acknowledge that the verifier introduces additional computational cost. However, we believe this cost is justified, as test-time scaling with a verifier provides valuable insights into the design of more powerful GUI models. Compared to prior grounding models such as UITARS and AGUVIS, **_GUI-Actor offers a key advantage: it can generate multiple candidate regions in a single forward pass, significantly reducing computation at this stage_**.
>
> To mitigate the cost introduced by the verifier, we **_have introduced an early stopping mechanism_** in our work—terminating verification once the confidence exceeds 0.95. This reduces the number of verifier calls substantially, as shown in **_Table G_**. On average, only 3.3 verifier calls are needed per sample on ScreenSpot-v2. Thanks to the efficient vLLM setup, this adds just 1.3 seconds per sample on average—an acceptable overhead with further room for parallelization and optimization. We also compared this approach with a **fully parallel solution** that scores all candidate areas simultaneously without early stopping. While it achieves similar latency, it incurs higher computational cost.
>
>
> **_Table G._** _Inference-time comparison of different grounding models with and without the Verifier._
> | Methods                                      | ScreenSpot-v2 | Average Verifier Calls (N) per sample | Time Consuming per sample in ScreenSpot-v2 | Estimated computational cost    |
> |---------------------------------------------|----------------|----------------------------------------|---------------------------------------------|----------------------------------|
> | baselines                                   | 86.0           | /                                     | 1.31s                                       | 1 * 7B                           |
> | Baseline w/ Verifier (Seqential Solution)                        | 88.0           | 9.9                                    | 21.22s                                      | 20 * 7B + N * 2B                 |
> | GUI-Actor w/o Verifier                      | 88.9           | /                                     | 1.42s                                       | 1 * 7B                           |
> | GUI-Actor w/ Verifier (Sequential Solution) | 89.5           | 3.3                                    | 2.72s                                       | 1 * 7B + N * 2B                  |
> | GUI-Actor w/ Verifier (Parallel Solution) | 89.7           | 20                                     | 3.03s                                      | 1 * 7B + 20 * 2B                 |
>
>
> **Q1:** _Is the self-attention applied to image-patch embeddings in the attention-based action head causal or non-causal? Could this choice affect performance?_
>
> The self-attention mechanism in the action head is **_non-causal_**. This design choice is intentional, as the goal is to enhance the model's ability to capture spatial relationships among image-patch embeddings. A non-causal setup **_allows each patch to attend to all others_**, which aligns with the intuition that **_spatial reasoning in GUI layouts requires global context rather than sequential or directional constraints_**. We believe this non-causal design is better suited for aggregating layout and UI structure effectively.
>
>
> **Q2:** _During GUI-Actor-7B’s supervised fine-tuning, which VLM parameters were actually trained?_
>
> During supervised fine-tuning of GUI-Actor-7B, we adopt a **_two-stage training_** strategy. First, we **_warm up the model by training only the newly introduced token embeddings and the parameters in the action head_**, while keeping the backbone VLM frozen. After this warm-up phase, we **_fine-tune the entire model_**, including the backbone VLM parameters. We will clarify this more explicitly in Line 221–224 in the revised manuscript.

---

> > ### Comment · Reviewer_8yA9 · 2025-08-02
> > **Response**
> >
> > Thank the authors for the detailed rebuttal and for addressing our concerns. I have the following comments:
> >
> > 1. W1: It’s suggested to add these training/design details in the paper for a better understanding. I have two further concerns:
> >
> > - The approach of injecting the \<ACTOR\> token directly into the input prompt (e.g., "pyautogui.click(\<ACTOR\>)") effectively hardcodes a specific action (click) and, as presented, seems to limit the method’s generality. It’s unclear how other actions, such as scroll, type-in, select, etc., would be supported under this design. Could the authors clarify whether the approach generalizes to a broader action space, or whether alternative formulations (e.g., having the model generate the \<ACTOR\> token instead of baking it into a fixed prompt) were considered?
> >
> > - The reply to W2 states that “the \<ACTOR\> token formatting is the same in both settings.” If the formatting is fixed in the same way even when the VLM weights are updated (e.g., GUI-Actor-7B with updated VLM), this raises the concern of introducing a reasoning bias: the model may over-rely on the prompt structure rather than learning to internally represent or infer action intent. Some discussion or ablation around this potential bias, and how it affects flexibility/reasoning, would strengthen the paper.
> >
> > As I was re-reading the manuscript while drafting this response, two additional observations emerged:
> >
> > 1. In both Tables 2 and 3, I found Actor-7B is not consistently outperforming GUI-Actor-7B w/ LiteTrain across all settings. Notably, in Table 2 (Mobile-Text, Mobile-Icon) and Table 3 (Avg-Text and especially CAD), there are cases where performance drops—CAD, in particular, shows a large decline (−9.2%) after fine-tuning the VLM. This is surprising and warrants deeper investigation. Why did it happen? Is it because of overfitting or degradation of certain representations of the original VLM? I’d encourage the authors to provide further analysis explaining why updating the VLM sometimes hurts performance, and specifically why the CAD setting degrades so sharply.
> >
> > 2. Regarding the real-world agent task, the authors curated an OSWorld-Windows subset for evaluation. Actor-7B w/o Verifier is constructed with GPT-4o as the planner and evaluated on 49 tasks, with 6 tasks completed successfully. Since this work mainly investigates the GUI grounding problem. It would be valuable to understand the failure modes more precisely: are most failures attributable to suboptimal planning from GPT-4o versus grounding errors from GUI-Actor variants?
> > Furthermore, since the ablation results show that GUI-Actor-7B w/ Verifier yields consistent gains. It would be more informative to see the analysis on the stronger model.

---

> > > ### Author Response · Authors · 2025-08-04
> > > **Thank you! We’ve addressed your questions as follows.**
> > >
> > > Thanks for the great questions! We address Q1 and Q2 together, as both raise points about the role of the `<ACTOR>` token.
> > >
> > > **Q1:** _injecting `<ACTOR>` token directly into input prompt...limit the method’s generality...unclear how other actions...be supported under this design...clarify whether the approach generalizes to a broader action space...(e.g., having the model generate the `<ACTOR>` token instead of baking it into a fixed prompt)...?_
> > >
> > > **Q2:** _..."`<ACTOR>` token...same in both settings." If the formatting is fixed...even when the VLM weights are updated...this raises the concern...model may over-rely on the prompt structure rather than learning to internally represent or infer action intent..._
> > >
> > > To clarify, **_our model is designed to support a broad range of actions_**, including click, scroll, write, select, etc., and **_even reasoning content_**, **_provided these actions are covered in training data_**. As the **_primary focus of this paper_** is on **_GUI grounding_** (following the same setting as UGround[1]), we explicitly **_injected the `<ACTOR>` token_** to evaluate the **_w/ LiteTrain model_**. This allows us to study the learning behavior of the newly introduced parameters in isolation. **_After the full fine-tuning stage, the model is fully capable of generating `<ACTOR>` tokens on its own_**. The `<ACTOR>` token serves as a trigger for the action head, while the specific action type (e.g., `pyautogui.click…` or `pyautogui.write…`) is flexibly produced by the model based on the context. For example, a generated output might look like:
> > > ```
> > > <think>To continue...homepage of Play Store.</think>
> > > Action: Click the back arrow.
> > > pyautogui.click(<ACTOR>)
> > > ```
> > > Or, depending on the context: `pyautogui.press("BACK")`, etc.
> > >
> > > We will include further discussion on generalizing across action types.
> > >
> > > **Q3:** _...Tables 2 and 3...GUI-Actor-7B is not consistently outperforming GUI-Actor-7B w/ LiteTrain...Why did it happen?...why updating the VLM sometimes hurts performance..._
> > >
> > > Thanks for pointing out the confusion. At first glance, the results in **_Tables 2 and 3_** may suggest that GUI-Actor-7B w/ LiteTrain sometimes matches or even outperforms the fully fine-tuned GUI-Actor-7B. However, this impression is primarily due to the **_enabling of the proposed Verifier_**, which provides an additional decision filtering and can substantially enhance performance of the main grounding model, particularly for the weaker LiteTrain variant.
> > >
> > > To clarify the performance of the core grounding model without the effects from the proposed Verifier, we present reorganized results below (**_Tables P, Q, and R_**, derived from **_Tables 4 and 5 in the main paper_**). These comparisons isolate **_grounding model performance without the Verifier_**, and reveal that **_GUI-Actor-7B consistently outperforms GUI-Actor-7B w/ LiteTrain across all benchmarks_**.
> > >
> > > **_Table P._** _Results (w/o Verifier) on ScreenSpot-Pro._
> > > | | Dev | Creative | CAD | Scientific | Office | OS | Avg-Text | Avg-Icon | Avg |
> > > |-|-|-|-|-|-|-|-|-|-|
> > > | GUI-Actor-7B | 38.8 | 37.2 | 23.0 | 40.2 | 57.0 | 34.2 | 52.6 | 14.7 | 38.1 |
> > > | GUI-Actor-7B w/ LiteTrain | 33.1 | 29.0 | 19.5 | 30.7 | 46.5 | 31.6 | 44.6 | 9.9 | 31.4 |
> > >
> > > **_Table Q._** _Results (w/o Verifier) on ScreenSpot and ScreenSpot-v2._
> > > | | Mobile-Text | Mobile-Icon | Desktop-Text | Desktop-Icon | Web-Text | Web-Icon | Avg |
> > > |-|-|-|-|-|-|-|-|
> > > |**_ScreenSpot_**|-|-|-|-|-|-|-|
> > > | GUI-Actor-7B | 93.8 | 78.6 | 92.8 | 81.4 | 90.9 | 79.1 | 86.6 |
> > > | GUI-Actor-7B w/ LiteTrain | 82.1 | 72.5 | 83.5 | 65.7 | 86.1 | 74.8 | 78.3 |
> > > |**_ScreenSpot-v2_**|-|-|-|-|-|-|-|
> > > | GUI-Actor-7B | 95.6 | 80.6 | 94.9 | 83.6 | 94.9 | 79.3 | 88.9 |
> > > | GUI-Actor-7B w/ LiteTrain | 84.5 | 79.6 | 87.2 | 69.1 | 90.7 | 72.2 | 81.7 |
> > >
> > > We will include these reorganized tables and a deeper analysis on how the Verifier helps in different domains in the revised paper.
> > >
> > > **Q4:** _...OSWorld-Windows...GUI-Actor-7B w/o Verifier is constructed with GPT-4o as the planner...with 6/49 tasks completed successfully...are most failures attributable to suboptimal planning from GPT-4o or grounding errors from GUI-Actor variants?...informative to see the analysis on the stronger model._
> > >
> > > We conducted a detailed **_analysis of the 43 failure cases_** from OSWorld-Windows. We found that the **_majority of failures (37 out of 43 cases, ~86%) are due to suboptimal GPT-4o planning_**. This observation is **_consistent with prior findings from UGround[1]_**: in tasks more aligned with the training data (e.g., MM-Mind2Web and AndroidControl-Low), ~90% of failures were attributed to Planner errors. In the detached planning + grounding framework, we believe **_the relative contribution of planning errors will become more prominent as the grounding component becomes more capable_**. We will include these analyses in the revised version.
> > >
> > >
> > > [1] Navigating the Digital World as Humans Do: Universal Visual Grounding for GUI Agents. ICLR 2025.

---

> > > > ### Comment · Reviewer_8yA9 · 2025-08-04
> > > > **Response**
> > > >
> > > > Thank authors for the response and it addresses most of my concerns. I’m particularly interested in the comparison between GUI-Actor-7B w/ LiteTrain and GUI-Actor-7B. From the description of the grounding verifier (L170–194), it appears to be actor-agnostic and trained separately from GUI-Actor-7B. In the ScreenSpot-Pro CAD setting, GUI-Actor-7B w/ LiteTrain exhibits worse standalone grounding performance than GUI-Actor-7B, yet after incorporating the verifier, its localization accuracy exceeds that of GUI-Actor-7B by 9.2%. Assuming the verifier is well trained, one plausible explanation is that fine-tuning GUI-Actor-7B degrades the recall of its grounding attention proposals relative to the LiteTrain variant, forcing the verifier to select from suboptimal candidates. It would be helpful if the authors could include an analysis of proposal recall for both methods to shed light on this behavior. Alternatively, they might consider other factors to investigate why it happens.

---

> ### Author Response · Authors · 2025-08-05
> **Thank you! We’ve addressed your follow-up questions.**
>
> Thank you for your thoughtful follow-up. Firstly, we performed a deeper analysis of the CAD examples and confirmed that this subset is particularly challenging. Most **_target UI elements in CAD_** are **_extremely small Chinese buttons_**, unlike **_other subsets_** that typically feature **_text-free icons or English text buttons_**. Given that our **_training data_** predominantly consists of **_English webpages_**, this domain gap may explain why the performance on the CAD subset is significantly lower than other subsets.
>
> To investigate further, we **_examined the attention maps_** produced by the **_LiteTrain and FullTrain_** (GUI-Actor-7B) models. While the **_LiteTrain_** model typically shows **_lower top-1 grounding accuracy_**, its **_attention maps are notably more dispersed_** compared to those from the FullTrain model, likely due to the frozen backbone VLM (including the ViT) in its training setup. **_This dispersion leads to more diverse region proposals being passed to the Verifier_**. Crucially, this **_improves proposal recall_**, i.e., the likelihood that the ground-truth region is among the top-k proposals.
>
> In contrast, the **_FullTrain_** model often produces **_sharper, more focused attention_**, which can be **_beneficial in general_** but may **_miss difficult or low-salience elements_** such as the small Chinese UI targets in CAD. As a result, LiteTrain—despite its weaker standalone grounding—can achieve superior final localization accuracy on CAD when combined with the Verifier.
>
> Additionally, we note that the **_underlying VLM_** backbone, Qwen2VL, was **_pretrained_** on a substantial amount of **_Chinese visual-language data_**. By freezing this backbone in **_LiteTrain_**, we **_preserve its stronger native capability in handling Chinese content_**. In contrast, full fine-tuning may inadvertently degrade some of these specialized representations, particularly when training is skewed toward English-centric data. This further contributes to LiteTrain’s relative advantage in the CAD setting.
>
> As suggested, we computed **_proposal recall_** (defined as the percentage of cases where the ground-truth region appears among the top-20 proposals) as shown in **_Table S_**. For comparison, we also include numbers from the Creative subset. The results further support our explanation:
>
> **_Table S._** _Proposal Recall on CAD and Creative subsets._
> |                  | LiteTrain | FullTrain |
> |------------------|-----------|-----------|
> | CAD Subset | 57.9  | 42.9  |
> | Creative Subset | 55.4  | 57.2  |
>
> We appreciate your suggestion and will include this analysis in a future revision to better characterize the interplay between attention diversity, language-specific capabilities, and Verifier effectiveness.

---

> > ### Comment · Reviewer_8yA9 · 2025-08-07
> > **Response**
> >
> > Thank authors for the response—the explanation is reasonable: the bias introduced by SFT data can indeed disrupt the original MLLM’s knowledge. If possible, I would be very interested to see whether, after incorporating additional downstream task–related data, GUI-Actor-7B can consistently outperform GUI-Actor-7B w/ LiteTrain in terms of proposal recall.

---

> > > ### Author Response · Authors · 2025-08-08
> > > **Thank you!**
> > >
> > > We are glad to hear that our explanation has improved the understanding of the observed behavior. Yes, incorporating additional downstream task–related data is an interesting direction to explore. As existing widely used open-source datasets lack Chinese-specific split, particularly in UI grounding scenarios, it would be important to first consider how to construct such a dataset. One possible approach would be to leverage VLM models to automatically analyze element size distributions and detect whether an example belongs to the Chinese domain, enabling targeted collection or augmentation of training samples. We believe such a dataset could help address the domain gap we identified and potentially allow GUI-Actor-7B to consistently outperform the LiteTrain variant in proposal recall. We will include any further findings along these lines in the revised version.

---

### Official Review · Reviewer_FWfD · 2025-07-01

**Clarity:** 3
**Significance:** 3
**Originality:** 3
**Rating:** 4
**Confidence:** 4

**Summary:**

GUI-Actor introduces an attention-based action head that uses a dedicated <ACTOR> token. This token directly focuses on relevant visual patch regions, removing the need for coordinate-based localization.

The method also handles spatial ambiguity by using multi-patch supervision. It labels all patches inside ground-truth bounding boxes as positives, ensuring better accuracy in GUI interactions.

Unlike traditional methods, GUI-Actor generates multiple candidates in a single forward pass. Older approaches require multiple inference runs with sampling.

**Questions:**

The attention-based action head design really needs more explanation. How does the self-attention layer actually pull together information from different patches? The paper talks about "GUI-aware contextual information" but doesn't say what makes it special for GUIs. Do they use custom positional encodings? Is there some way to model how GUI elements relate to each other spatially?

The multi-patch supervision approach could use some ablation studies too. When they decide which patches count as "positive" based on overlap, how much does performance change if you tweak those thresholds? It seems like this could make a big difference but they don't test it.

**Ethical Concerns:**

["NO or VERY MINOR ethics concerns only"]

**Final Justification:**

I raised concerns about the attention-based action head design and multi-patch supervision approach. The authors provided detailed responses. They explained the architecture specifics including 8 attention heads and two-layer MLPs. They clarified how self-attention aggregates spatial relationships.They described their negative sampling strategy that includes semantically meaningful UI elements. They also provided an error analysis table examining failure modes. These responses reasonably address my questions. I would like to increase my scores for this work.

**Limitations:**

Yes.

**Quality:**

3

**Strengths And Weaknesses:**

Strengths:

GUI-Actor ditches coordinate generation entirely. Instead, it uses an attention-based action head with a special <ACTOR> token. This creates a direct connection between what the model sees and what it wants to do. No need for text-based coordinate prediction.

The framework can generate multiple candidate regions in one go. It does this through attention maps from the transformer backbone. This beats baseline methods that have to run inference multiple times just to get candidates.

GUI-Actor can also handles new scenarios. It performs great on out-of-distribution benchmarks like ScreenSpot-Pro. The attention mechanism adapts to different screen resolutions automatically. It deals with varying layouts much better than coordinate-based methods that struggle with resolution changes.

Weaknesses:

The paper doesn't really explain how the attention-based action head works. Key parts like the "GUI-aware" self-attention are mentioned but not detailed. We don't know how many attention heads are used or how they're configured. Same goes for the MLP projection designs. The exact architecture of the action head remains unclear. This makes it hard to understand or reproduce.

The experimental setup has some issues. The verifier training uses simple binary classification with only correct/incorrect labels. It relies on fake negative examples generated by random coordinate shifts. But real problems are usually messier than that. Users might click slightly off-target or have different interaction patterns.

There's also a lack of thorough error analysis. The authors don't dig into when and why things go wrong. What types of GUI elements cause failures? The verifier training uses those oversimplified artificial negatives.

---

> ### Author Rebuttal · Authors · 2025-07-31
>
> Thank you for reviewing our work and providing valuable feedback. We have carefully addressed your concerns below. Please let us know if you have any further questions.
>
> We address W1 and Q1 together, as they both raise points about the architecture and design of the attention-based action head.
>
> **W1:** *The paper doesn't really explain how the attention-based action head works. Key parts like the "GUI-aware" self-attention are mentioned but not detailed. We don't know how many attention heads are used or how they're configured. Same goes for the MLP projection designs. The exact architecture of the action head remains unclear. This makes it hard to understand or reproduce.*
>
> **Q1:** *The attention-based action head design really needs more explanation. How does the self-attention layer actually pull together information from different patches? The paper talks about "GUI-aware contextual information" but doesn't say what makes it special for GUIs. Do they use custom positional encodings? Is there some way to model how GUI elements relate to each other spatially?*
>
> Thanks for pointing out the confusion. As shown in Figure 2, Eq (2), Eq (3) and described in Line 150-152, the action head comprises three main components: **_(1) a multi-head self-attention layer, followed by (2) an MLP to extract vision patch features, and (3) a [ACTOR] token-specific MLP to extract semantic features_**.
>
>
>
> The **_self-attention layer_** operates over image patches, where **_each patch representation is computed as an attention-weighted sum of all other patch representations_**. This mechanism allows the model to **_explicitly aggregate spatial relationships_** among UI elements and layout structures beyond what the original ViT encoder captures, by leveraging multi-patch supervision.
>
> Regarding positional encoding, we do **_not introduce additional positional embeddings in this self-attention layer_**, as the input features from the ViT backbone already incorporate positional information.
>
> For the **_configuration details_**: The number of attention heads in the self-attention layer is set to 8; Both MLP components are two-layer feedforward networks with a GELU activation in between. As stated in Line 213–214, we use the same dimensionality as the backbone VLM for all dimension configuration within the action head.
>
> We will revise the manuscripts and open source our code to ensure clarity and reproducibility.
>
>
> **W2:** *The experimental setup has some issues. The verifier training uses simple binary classification with only correct/incorrect labels. It relies on fake negative examples generated by random coordinate shifts. But real problems are usually messier than that. Users might click slightly off-target or have different interaction patterns.*
>
> Thank you for the feedback. We believe the comment may stem from a partial interpretation of our setup, and we would like to clarify the rationale and design behind our verifier training. **(1) Use of Constructed Negatives:**  While the negative examples are indeed constructed rather than collected from real user interactions, **_this contrastive strategy is a widely used and effective practice in many learning paradigms_** such as contrastive learning and self-supervised learning. In our case, these constructed negatives help the verifier distinguish correct clicks from plausible but incorrect alternatives. **(2) Diversity of Negative Samples:** Our negative sampling is not limited to simple random coordinate shifts. As detailed in **Line 185-187**, we deliberately introduced varying levels of difficulty_** to better reflect real-world behavior. In addition to **_random offsets (which naturally include slightly off-target cases)_**, we also **_select negative points located within the bounding boxes of other semantically meaningful UI elements_**. This design simulates realistic interaction ambiguities, such as users clicking on relevant but incorrect elements.
>
>
> **W3:** *There's also a lack of thorough error analysis. The authors don't dig into when and why things go wrong. What types of GUI elements cause failures? The verifier training uses those oversimplified artificial negatives.*
>
> Thank you for the valuable suggestion. We conducted a **_detailed error analysis_** across 40 examples from SS-Pro and SS-v2, respectively. **_Table E_** shows the details. We hope this breakdown offers insight into when and why failures occur and shows how dataset properties (e.g., ambiguity, element size) affect performance. We will include this analysis in the next version to guide future improvement.
>
> **_Table E._** _Error Analysis on 40 examples from SS-pro and SS-v2, respectively._
> | Error Type                                                                                             | ScreenSpot-Pro  | ScreenSpot-v2   |
> |--------------------------------------------------------------------------------------------------------|---------|---------|
> | The ground-truth valid action region is not unique (model captures the one not labeled as ground-truth). | 7.50%   | 7.50%   |
> | The instruction is ambiguous (models take reasonable but not accurate actions given the non-specified instruction). | 7.50%   | 20.00%  |
> | The ground-truth region is smaller than 14px * 14px.                                                   | 5.00%   | 0.00%   |
> | The ground-truth region gains some attention, but fails to produce a final accurate position.          | 47.50%  | 27.50%  |
> | The ground-truth region gains little, even no attention.                                               | 32.50%  | 45.00%  |
>
> Please refer to W2 for the concern on verifier training with artificial negatives.
>
>
> **Q2:** *The multi-patch supervision approach could use some ablation studies too. When they decide which patches count as "positive" based on overlap, how much does performance change if you tweak those thresholds? It seems like this could make a big difference but they don't test it.*
>
> The **_multi-patch supervision during training is determined objectively_**. As described in Line163-164 and illustrated in Figure 2b, our multi-patch supervision strategy labels all image patches that are partially or fully covered by the ground-truth bounding box as positive (label = 1), and all other patches as negative (label = 0). This **_binary supervision is applied objectively based on spatial overlap, without threshold tuning_**. While it is possible to apply soft labels to boundary patches (e.g., based on intersection ratios), we conjecture the impact would be limited, as only a narrow band of edge patches would be affected. We will consider this in future work.
>
> Regarding the **_threshold used at inference time_** to aggregate multiple clustered patches into a final prediction, we conducted an ablation by varying the value from 0.1 to 1.0. As shown in **_Table F_**, performance remains stable within a moderate range (0.3 to 0.6). However, extreme values, _i.e.,_ either too permissive (0.1) or too restrictive (1.0, equivalent to using only the top-1 patch), lead to drops in accuracy. These results indicate that while the threshold choice does impact performance, **_the method is robust across a reasonable range_**. We will include this ablation in the final version to clarify this design choice.
>
> **_Table F._** _Ablation study on the attention threshold for final prediction. Numbers from 0.3 is reported in the paper._
> | Dataset         | 0.1  | 0.2  | 0.3 | 0.4  | 0.5  | 0.6  | 0.7  | 0.8  | 0.9  | 1.0 (Top-1 Patch Center) |
> |----------------|------|-----|------|------|------|------|------|------|------|--------------|
> | ScreenSpot-v2  | 87.6 | 88.8 | 88.9 | 88.4 | 88.1 | 86.7 | 86.2 | 84.1 | 83.2 | 81.5         |
> | ScreenSpot-Pro | 34.5 | 36.4 | 38.1 | 37.8 | 38.6 | 38.0 | 35.7 | 32.7 | 30.0 | 27.4         |

---

> > ### Comment · Reviewer_FWfD · 2025-08-08
> > **Response to Authors' Rebuttal**
> >
> > Thank you for the authors' feedback, which has addressed my concerns to some extent. Based on these responses, I would like to revise my scores upward.

---

> > > ### Author Response · Authors · 2025-08-09
> > > **Thank you!**
> > >
> > > Thank you for your time and effort in reviewing our work!

---

### Official Review · Reviewer_LxDz · 2025-07-02

**Clarity:** 4
**Significance:** 4
**Originality:** 4
**Rating:** 5
**Confidence:** 4

**Summary:**

This paper introduces GUI-Actor, a new framework for Vision-Language Model (VLM) based GUI agents that overcomes the limitations of traditional text-based coordinate generation. The authors argue that relying on coordinates creates issues with spatial-semantic alignment, action target ambiguity, and mismatched granularity between vision and screen. GUI-Actor offers a coordinate-free solution. It uses a special <ACTOR> token and a unique "action head" to directly generate an attention map over a screenshot, pinpointing the most relevant area for an action. To handle the ambiguity of target locations (e.g., any part of a button), the model is trained using a multi-patch supervision method where any visual patch overlapping with the correct action area is considered a valid target. A lightweight "grounding verifier" then refines the selection from high-attention regions. Experiments show GUI-Actor significantly surpasses existing methods, especially in generalizing to new, high-resolution applications.

**Questions:**

1. The paper identifies the limitation regarding small UI elements due to fixed patch sizes. Could you provide a more quantitative analysis of this issue? For instance, what proportion of errors on ScreenSpot-Pro are attributable to this, and how effective is the described clustering refinement strategy at mitigating them? Have you considered architectural changes to address this more directly, such as incorporating multi-scale features from the vision encoder into the action head?

2. The verifier is shown to be general enough to boost the performance of the AGUVIS baseline. This is a strong result. Given that your coordinate-free method excels at general localization and coordinate-based methods can offer high precision, have you considered a hybrid model? For example, one that uses the attention head to identify the correct UI element and a secondary mechanism to predict a precise click coordinate within that element's bounding box.

3. The training data for the verifier is constructed from the OS-Atlas dataset, while the main model uses a different collection of datasets. Could you discuss the potential impact of this domain shift? Would there be a benefit to training the verifier on the exact same data distribution that the main model is trained on?

**Ethical Concerns:**

["NO or VERY MINOR ethics concerns only"]

**Final Justification:**

Thank you to the authors for the detailed rebuttal. I appreciate the clear explanations and the additional experimental results. All of my previous concerns have been adequately addressed.

**Limitations:**

Yes

**Paper Formatting Concerns:**

No paper formatting issues are found in this paper.

**Quality:**

4

**Strengths And Weaknesses:**

Strengths:

- The work tackles a well-defined and critical bottleneck in the development of capable GUI agents. Moving away from brittle coordinate generation is a crucial step for improving the robustness and generalization of these systems.

- The paper's methodology and evaluation are of excellent quality. The proposed architecture is well-motivated and technically sound. The experimental setup is comprehensive, featuring multiple standard benchmarks, rigorous comparisons against relevant baselines.

- The paper is exceptionally well-written, structured, and easy to follow.

- The core idea of reformulating GUI grounding as a coordinate-free, attention-based localization task is novel and insightful within this specific domain.

Weaknesses:

- The primary weakness, which the authors acknowledge, is the dependency on the backbone VLM's fixed patch size. This makes it difficult to ground actions on very small UI elements that may be smaller than a single visual patch.

---

> ### Author Rebuttal · Authors · 2025-07-31
>
> Thank you for reviewing our work and providing valuable feedback. We have carefully addressed your concerns below. Please let us know if you have any further questions.
>
> **W:** *The primary weakness, which the authors acknowledge, is the dependency on the backbone VLM's fixed patch size. This makes it difficult to ground actions on very small UI elements that may be smaller than a single visual patch.*
>
> In our proposed method, the attention-weighted center of top-K patches can help mitigate this issue and localize small UI elements (**_<14×14 pixels_**) that overlap with 3-4 patches. However, current design of GUI-Actor may still struggle when such elements **_fall entirely within a single patch_**. Fortunately, these extreme cases are **_relatively rare_** in typical computer use scenarios, as shown in **_Table A_**.
>
> To mitigate this limitation, we explored a simple yet effective **_"Zoom-In" strategy_**, where GUI-Actor makes an initial coarse prediction, then zooms in 2× around the predicted location and performs a second round of grounding. **_Table B_** shows this significantly boosts performance on ScreenSpot-Pro, demonstrating its effectiveness in mitigating grounding errors in high-resolution settings with small UI elements.
>
> **_Table A._** _Frequency of small UI elements (longest edge < 14 px) fully contained within a single patch._
> | Dataset         | # of Examples | # of with Small Elements | # of Small Element In One Patch |
> |-----------------|----------------------|--------------| --------------|
> | ScreenSpot-v2    | 1272              | 0                    | 0            |
> | ScreenSpot-pro   | 1581              | 52                   | 10            |
>
> **_Table B._** _Effectiveness of the "Zoom-In" strategy on ScreenSpot-Pro._
> | Model for Zoom-In Round              | Model for Final Round              | Zoom-in Ratio | Screenspot-Pro |
> |-------------------------------------|------------------------------------|----------------|----------------|
> | /                                   | Aguvis-7B                          | /              | 22.9           |
> | /                                   | GUI-Actor-7B                       | /              | 38.1           |
> | GUI-Actor-7B (w/o Verifier)         | Aguvis-7B                          | 2x             | 43.0           |
> | GUI-Actor-7B (w/o Verifier)         | GUI-Actor-7B (w/o Verifier)        | 2x             | **50.28**      |
>
>
> **Q1:** *The paper identifies the limitation regarding small UI elements due to fixed patch sizes. Could you provide a more quantitative analysis of this issue? For instance, what proportion of errors on ScreenSpot-Pro are attributable to this, and how effective is the described clustering refinement strategy at mitigating them? Have you considered architectural changes to address this more directly, such as incorporating multi-scale features from the vision encoder into the action head?*
>
> Thank you for the constructive suggestion. We provide a more detailed quantitative analysis in **_Table C_**, where ScreenSpot-Pro targets are grouped by bounding box area relative to the patch area (**_n = 14×14_**). As shown, accuracy for the smallest elements $[0,n)$ is significantly lower than for larger ones, underscoring the challenge of grounding small UI components. Notably, Table C also shows that our **_proposed refinement strategy  (attention-weighted center) consistently improves accuracy across all size ranges_**, including the most difficult small-target cases.
>
> We appreciate the suggestion regarding architectural improvements. Integrating multi-scale features,  coarse-to-fine attention, or learning an offset is a promising direction for improving fine-grained localization. We leave this for future work.
>
> **_Table C._** _Accuracy by target UI element size on ScreenSpot-Pro, measured by area, n=14*14, without Verifier._
> | Range      | # of Examples | Accuracy of Top-1 Patch Center | Accuracy of Weighted Sum of Patch Cluster|
> |------------|----------------|-------------------------------|------------------------------------------|
> | [0, n)     | 23             | 4.35%                         | 13.04% (↑ 200%)                          |
> | [n, 4n)    | 349            | 6.59%                         | 11.46% (↑ 73.91%)                        |
> | [4n, 9n)   | 367            | 14.99%                        | 25.07% (↑ 67.27%)                        |
> | [9n, )     | 842            | 42.04%                        | 55.58% (↑ 32.20%)                        |
> | All        | 1581           | 27.39%                        | 38.14% (↑ 39.26%)                        |
>
>
> **Q2:** *The verifier is shown to be general enough to boost the performance of the AGUVIS baseline. This is a strong result. Given that your coordinate-free method excels at general localization and coordinate-based methods can offer high precision, have you considered a hybrid model? For example, one that uses the attention head to identify the correct UI element and a secondary mechanism to predict a precise click coordinate within that element's bounding box.*
>
> Thank you for the insightful question. **_Table D_** presents the results for a two-stage strategy in which the original screenshot is **_first cropped to half its size_**  (H x W -> H/2 x W/2), **_centering the first-round prediction_**. Then the cropped image is used as input for a **_second-round prediction_**. This setup can be viewed as a **_hybrid model_** when different models are used across the two stages.
>
> As shown in **_Table D_**, this two-stage approach consistently improves performance. It demonstrates that **_GUI-Actor not only offers more accurate coarse-grained localization but is also capable of producing finer-grained click coordinates_**. While **_small UI elements_** pose challenges for GUI-Actor in predicting precise coordinates, they **_similarly affect coordinate generation-based methods_**. In cases where UI elements are very small, coordinate-based models may struggle to capture and learn the correct semantics without explicit spatial alignment. In contrast, GUI-Actor benefits from such explicit alignment through its action head, which helps guide more precise localization.
>
> **_Table D_.** _Effectiveness of the hybrid strategy on ScreenSpot-Pro._
> | Model for First Stage (Crop)     | Model For Second Stage              | Crop Ratio | Screenspot-Pro |
> |----------------------------------|-------------------------------------|------------|----------------|
> | Aguvis-7B                        |                                     | /          | 22.9           |
> | GUI-Actor-7B                     |                                     | /          | 38.1           |
> | Aguvis-7B                        | Aguvis-7B                           | 1/2        | 41.6          |
> | GUI-Actor-7B (w/o Verifier)      | Aguvis-7B                           | 1/2        | 44.4          |
> | Aguvis-7B                        | GUI-Actor-7B (w/o Verifier)         | 1/2        | 46.1          |
> | GUI-Actor-7B (w/o Verifier)      | GUI-Actor-7B (w/o Verifier)         | 1/2        | **46.3**      |
>
>
> **Q3**: *The training data for the verifier is constructed from the OS-Atlas dataset, while the main model uses a different collection of datasets. Could you discuss the potential impact of this domain shift? Would there be a benefit to training the verifier on the exact same data distribution that the main model is trained on?*
>
> Thank you for the question.  **_Our Verifier was trained during an earlier stage_** of the project using the OS-Atlas dataset, before the UGround bounding-box dataset was released on May 1, 2025 (available on Hugging Face). Generating a full UGround-based verifier dataset requires extra time, so we decided not to create a new verifier dataset and train it again. Thanks to our diverse negative-sampling strategy and the inherent variety in OS-Atlas, **_the Verifier generalizes very well_**—even when paired with a grounding model trained on different datasets. While one could certainly retrain the Verifier on the exact same distribution as the main model, our experiments suggest that the existing setup already delivers strong performance with minimal domain-shift impact.

---

### Note · Authors · 2025-08-14

Dear AC and Reviewers,

We sincerely appreciate your effort in overseeing the review process. As the rebuttal phase concludes, we would like to provide a concise summary of our discussions and responses.

**Notable strengths identified in reviews**
- The problem is well-defined and addresses a crucial bottleneck in developing GUI agents (LxDz).
- The idea of exploring attention-based grounding—rather than brittle coordinate generation—is novel, insightful, and conceptually sound (LxDz, 8yA9, YmmE). This approach represents an important step toward improving robustness and generalization (LxDz).
- The proposed methodology and architecture is novel (YmmE), well-motivated and technically sound (LxDz). Introducing an `< ACTOR >` token as a grounding anchor for computing attention with image patch embeddings is a good design choice (8yA9). It creates a direct connection between what the model sees and what it wants to do, and it can generate multiple candidate regions in one go, beating baseline methods that have to inference multiple times (FWfD).
- The evaluation is of excellent quality with comprehensive experimental setup, rigorous comparisons against baselines (LxDz), thorough ablations (YmmE), and strong OoD results (FWfD).
- The paper is clear, well-organized, and easy to follow. (LxDz, 8yA9, YmmE)

**Key clarifications and additions during rebuttal**
- Added statistical analyses, “Zoom-In” experiments, and “Crop” experiments for locating small UI elements on ScreenSpot-Pro (LxDz, YmmE).
- Clarified training data construction and the inference mechanism of the Grounding Verifier (LxDz, FWfD, YmmE).
- Provided detailed design choices and hyperparameters for the proposed action head, along with experiments on hyperparameter selection (FWfD, 8yA9).
- Conducted error analysis on ScreenSpot-Pro, ScreenSpot-v2, and OSWorld-Windows (FWfD, 8yA9).
- Clarified training and inference mechanisms of the main grounding model and its LiteTrain variant, and provided intermediate results on proposal recall (8yA9).
- Added analysis and latency evaluation of the Grounding Verifier (8yA9, YmmE).
- Reorganized key results and extended the evaluation from Qwen2VL as the backbone VLM to Qwen2.5VL and UI-TARS-7B as backbone VLMs (YmmE).

These clarifications were well received by all reviewers, who expressed satisfaction with the rebuttal and the additional analyses provided.

We thank you again for the constructive discussions, which have helped strengthen and refine our work.

---

### Decision · Program_Chairs · 2025-09-17

**Decision:**

Accept (poster)

**Comment:**

This paper presents GUI-Actor, a coordinate‑free attention-based grounding framework for GUI agents. The approach is novel within GUI grounding, well motivated, clearly described, and extensively validated across ScreenSpot, ScreenSpot‑v2, ScreenSpot‑Pro and additional backbones, with strong OoD generalization and thorough ablations. Rebuttal and follow‑up clarifications satisfactorily addressed concerns on small targets, architecture details, efficiency, and fairness. All reviewers converge on positive recommendations. The AC agrees and recommends Acceptance.